# Universal logic with encoded spin qubits in silicon

Aaron J. Weinstein[1✉], Matthew D. Reed[1], Aaron M. Jones[1], Reed W. Andrews[1], David Barnes[1], Jacob Z. Blumoff[1], Larken E. Euliss[1], Kevin Eng[1], Bryan H. Fong[1], Sieu D. Ha[1], Daniel R. Hulbert[1], Clayton A. C. Jackson[1], Michael Jura[1], Tyler E. Keating[1], Joseph Kerckhoff[1], Andrey A. Kiselev[1], Justine Matten[1], Golam Sabbir[1], Aaron Smith[1], Jeffrey Wright[1], Matthew T. Rakher[1], Thaddeus D. Ladd[1✉] & Matthew G. Borselli[1]

Quantum computation features known examples of hardware acceleration for certain problems, but is challenging to realize because of its susceptibility to small errors from noise or imperfect control. The principles of fault tolerance may enable computational acceleration with imperfect hardware, but they place strict requirements on the character and correlation of errors[1]. For many qubit technologies[2–21], some challenges to achieving fault tolerance can be traced to correlated errors arising from the need to control qubits by injecting microwave energy matching qubit resonances. Here we demonstrate an alternative approach to quantum computation that uses energy-degenerate encoded qubit states controlled by nearest-neighbour contact interactions that partially swap the spin states of electrons with those of their neighbours. Calibrated sequences of such partial swaps, implemented using only voltage pulses, allow universal quantum control while bypassing microwave-associated correlated error sources[1,22–28]. We use an array of six $^{28}$Si/SiGe quantum dots, built using a platform that is capable of extending in two dimensions following processes used in conventional microelectronics[29]. We quantify the operational fidelity of universal control of two encoded qubits using interleaved randomized benchmarking[30], finding a fidelity of 96.3% ± 0.7% for encoded controlled NOT operations and 99.3% ± 0.5% for encoded SWAP. The quantum coherence offered by enriched silicon[5–9,16,18,20,22,27,29,31–37], the all-electrical and low-crosstalk-control of partial swap operations[1,22–28] and the configurable insensitivity of our encoding to certain error sources[28,33,34,38] all combine to offer a strong pathway towards scalable fault tolerance and computational advantage.

The ultimate requirements for useful quantum hardware are set by fault tolerance (FT), whereby information is encoded in a way that contains and negates errors with a combination of redundancy, symmetry and careful scheduling of operations. FT requires, in part, that qubits be well-isolated from microscopic sources of noise and controlled with precision and high speed, all in a platform capable of scaling to sizes of computational relevance. Achieving the necessary scale favours lithographically defined qubits such as superconducting transmons[39] or single electron spins in Si quantum dots[22]. These approaches have enjoyed significant recent progress in scaling[3,40], control fidelity[5–7,41] and advanced fabrication[42,43]. Crucially, however, FT also depends sensitively on the structure and correlation of the errors it is responsible for mitigating. Accordingly, qubit platforms that seem to satisfy the scale and fidelity requirements of quantum algorithms could nevertheless fail to achieve FT and so too fall short of computational acceleration. One significant source of possible correlated noise results from finite and varying energy splittings between qubit states, featured in most single-spin and transmon qubit systems. The phase evolution set by those splittings must be stable and continuously tracked, and conventional methods of control based on resonant driving between such splitting present challenges with respect to crosstalk.

Here we report on the realization of an alternative approach to quantum computation (QC) that uses degenerate qubit states and coherent partial swap interactions, the combination of which intrinsically avoids the FT challenges related to finite qubit energy splittings. Conventionally, qubits are controlled with arbitrary unitary rotations such as $R_x(\theta) = \cos(\theta/2)\,\mathrm{ID} + i\sin(\theta/2)\,\sigma_x$, where ID is the identity and $\sigma_x$ is the quantum analogue of the classical NOT, associated with adding or removing energy from a qubit. The combination of $R_v(\theta)$ (for arbitrary axis **v** and angle $\theta$) and a two-qubit gate such as controlled NOT (CNOT) constitutes a universal quantum gate set[44], analogous to the universal NOT and AND gates for classical bits (Fig. 1). This quantum gate set is able to traverse the very large continuous special unitary (SU) group of $SU(2^n)$, where $n$ is the number of qubits, rather than the (still very large) discrete symmetric group of $2^n$ bitstrings in the case of classical digital computers. Our alternative method of qubit control evokes the

[1]HRL Laboratories, LLC, Malibu, CA, USA. ✉e-mail: ajweinstein@hrl.com; tdladd@hrl.com

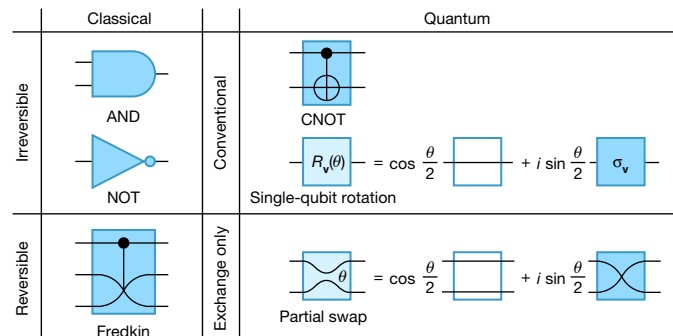

**Fig. 1 | Example universal gate sets for four computational models.** Top left, digital Boolean logic is universal with NOT and AND. Top right, single-spin-based quantum, with rotations about vector **v**, $R_v(\theta)$ and CNOT. Bottom left, all Boolean logic may be performed reversibly within a subsystem using the Fredkin gate. Likewise, bottom right, all quantum logic may be performed within a subsystem using partial swaps.

classical 'conservative logic'[45] of controlled-swapping of bits, which, as a Hamming-weight-conserving operation, operates on a smaller set of bitstrings scaling as $2^n/n$ for large $n$. This logarithmic overhead in bit number may be an acceptable cost for the benefit, in principle, of reducing heat associated with bit flips. In our quantum case, we apply arbitrary unitary superpositions of swapping and not swapping spins $j$ and $k$ of an $n$-spin array, that is, 'partial swaps', written as $U_{jk}(\theta) = \cos(\theta/2)$ ID $+ i\sin(\theta/2)$ SWAP$_{jk}$ for any angle $\theta$. Such a quantum computer could traverse SU($\approx 2^n/n^{3/2}$) for large $n$ (ref. [23]).

Our demonstration uses a further refined subsystem, encoding $n/3$ qubit states in $n$ spins. The remaining spin states outside this encoding are called leakage states, and their availability using controlled partial swapping provides a necessary resource for multi-qubit encoded universal quantum logic[1,24]. The linearly reduced qubit number and control-complexity overhead of our encoding relative to single-spin-based qubits enables the storage and control of qubit information in gapless quantum states. Indeed, this embedding was

first theoretically proposed more than 20 years ago[1,23–25] precisely to avoid the correlated error whereby a global magnetic field fluctuation could dephase every qubit in a single-spin-based quantum computer at once. Such highly correlated errors would be ruinous to FT. By using only partial swaps in a global uniform field, this type of correlated error is avoided entirely; such encodings were hence called noiseless codes[25] or decoherence-free subsystems (DFSs)[1].

We implement the partial swap operation needed for DFS qubits with the calibrated activation of the contact exchange interaction between lithographically defined quantum dots. This highly local interaction, which arises as a consequence of the Pauli exclusion principle, is also commonly used for two-qubit operations in single-spin qubits and in double-spin singlet-triplet qubits[22] in conjunction with mechanisms to differentiate spin-splitting energies (for example, magnetic field gradients). For exchange-only (EO) operation, however, a critical design difference is the choice to remove as much variation in spin splittings as possible to maintain encoded-state degeneracy. EO operation has been demonstrated previously in GaAs[26] and Si-based[27,35] triple-quantum dots, but those encodings were limited to a single qubit with no ability to use leakage states to complete encoded universal quantum logic[1,23].

Our device confines electrons in a 5-nm-thick silicon well that is isotopically enhanced to a residual 800-ppm content of $^{29}$Si; the well is surrounded by isotopically natural Si$_{0.3}$Ge$_{0.7}$ barriers. Isotopic enhancement reduces gradients in the effective magnetic field seen by electron spins because of hyperfine interactions[22], which constitute the dominant source of undesired spin-splitting variation and resulting control error for our encoding. Lithographically patterned metallic gates with tuned biases establish the electrostatic potentials that laterally confine and control interactions between electrons in the six quantum dots comprising the qubits and the two measurement dots (Fig. 2a,b). Voltages on plunger gates (P1–P6) accumulate a single electron in each dot and exchange gates (X1–X5) control the interaction strength between neighbouring electrons. Extra tunnel (T1, T6), bath (B1–B3) and charge sensor gates (M1, M2, Z1–Z4) (Fig. 2a) enable initialization and readout[46].

Our previous work[32,33,35,46–48], similar to other recent silicon qubit results[5,6,8,9,49], used an architecture that uses overlapping aluminium

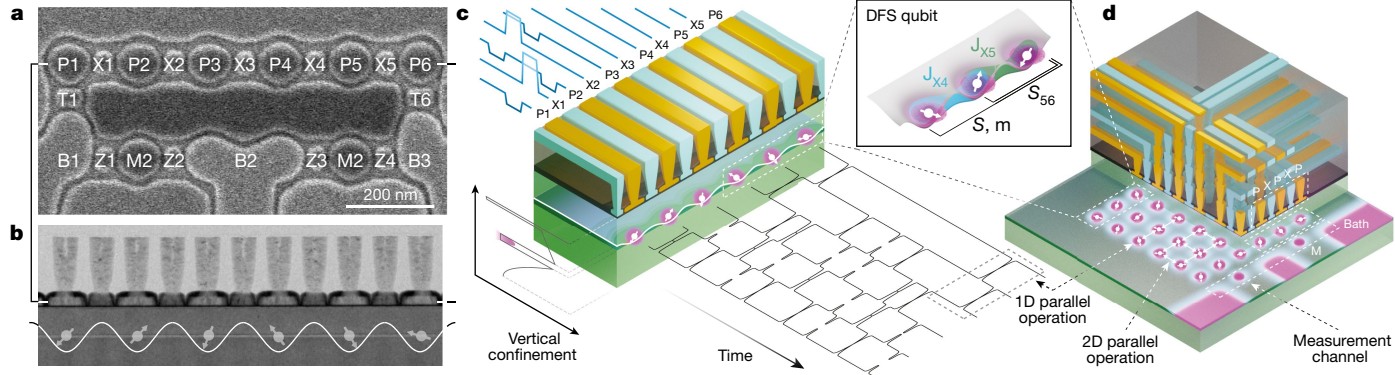

**Fig. 2 | A six-quantum-dot, two-qubit SLEDGE device in silicon. a**, Top-view scanning electron micrograph of the etched TiN single metal layer, defining the device's plunger (P), exchange (X), tunnel (T), bath (B), barrier (Z) and measure-dot (M) electrode gates. Two DFS qubits are formed with the P1–P3 and P4–P6 dots and connected by a single exchange gate (X3). Unlabelled patches of metal are field gates, held at constant bias. **b**, Cross-sectional transmission electron micrograph of gate and TiN via electrodes, cut along the line of plunger and exchange gates in (**a**). Electrons depicted as circles with arrows are vertically confined by the Si/SiGe heterostructure boundaries and laterally confined by the induced electrostatic potential from the device gates. Al$_2$O$_3$/HfO$_2$ dielectrics appear as a black boundary around the electrode gates[29].

**c**, Qubit states are manipulated using sequences of voltage pulses ('playing' from the upper left in this schematic) that actuate nearest-neighbour exchange interactions. The interactions are principally modulated by X gates[33,35]. We perform entangling operations on the two-qubit encoded subspace with sequences of partial swaps such as the sequence shown playing to the lower right. **d**, Schematic of a future multilayer-BEOL SLEDGE device that enables scaling in the lateral dimension. Devices designed using this capability could feature several rails of coupled quantum dots and readout channels by leveraging the standard methods of BEOL signal interconnection used in the semiconductor microelectronics industry.

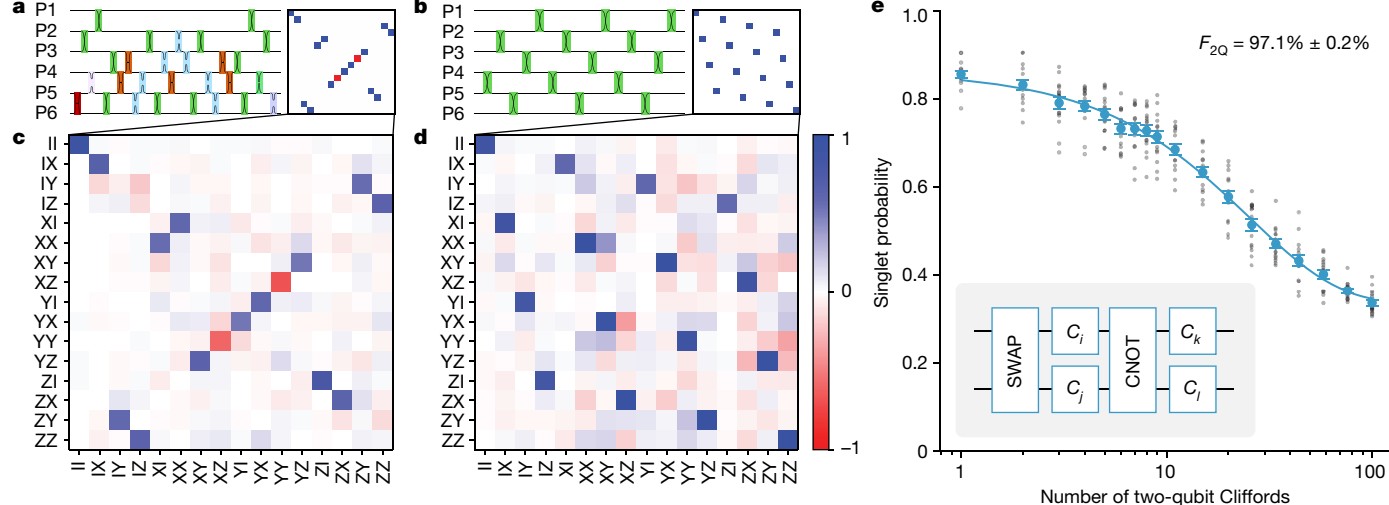

**Fig. 3 | Two-qubit process tomography and RB. a,b**, Exchange pulse diagrams and ideal process matrices of FW-CNOT (**a**) and SWAP (**b**) gates. The shading of exchange pulse boxes is proportional to the pulse exchange angle; detailed pulse angles may be found in Extended Data Tables 1 and 2. **c,d**, Maximum-likelihood estimates of measured quantum process matrices of FW-CNOT and SWAP gates. These data depend on joint measurement, so the relatively low M2 SPAM fidelity reduces contrast. The CNOT tomogram was measured on a similar device from the same wafer, with the same well width and gate pitch and comparable SPAM. **e**, Two-qubit RB with $t_{pulse} = 10$ ns, $t_{idle} = 5$ ns and applied magnetic field $B = 2.1$ mT. Individual dots represent the average from 500 shots of a single instance of a random Clifford sequence. Error bars here represent the $1\sigma$ standard deviation calculated at each Clifford length. Two-qubit Clifford gates are compiled using FW-CNOT, SWAP and single-qubit Clifford gates $C_x$ (inset).

gates, which was challenging to reliably yield performant quantum dots at the level required for practical EO operation in extended arrays. Our present demonstration is instead based on the single-layer etch-defined gate electrode (SLEDGE) platform, recently detailed in Ha et al.[29], whose improved yield results from separation in design and fabrication of the gate-defining layer (the 'single layer', which is a sheet of TiN-on-oxide above the silicon) from the back-end-of-line (BEOL) wiring, which contacts the gate layer through vertical via structures, as seen in Fig. 2. Although our device contains only two rows of quantum dots, the SLEDGE architecture readily allows extension of this geometry to two-dimensional (2D) arrays by stacking via-connected routing layers, as shown in Fig. 2d, following the BEOL interconnect strategies of modern silicon microprocessors[29]. Existing demonstrations of 2D arrays of semiconductor qubits use either carrier depletion in GaAs[10] or ring-like geometries of overlapping gates[11,12], both posing challenges to scaling that are circumvented by the via-based approach of SLEDGE. Arbitrarily large 2D arrays are still prevented by a finite number of feasible routing layers at the quantum dot pitch, but sufficient connectivity should nevertheless be available to enable 2D codes[13,14,50] when using the high-fidelity native encoded-SWAP operation that we demonstrate below.

We encode qubits using spins capable of partial swaps by associating quantum information with the total angular momentum quantum number of pairs of two spins, say 1 and 2, where states with $S_{12} = 0$ (the antisymmetric singlet state) map to $|0\rangle$ and states with $S_{12} = 1$ (the symmetric triplet state) map to $|1\rangle$. Hence $S_{12}$ and $S_{56}$ represent the two qubit states in our six-dot array (Fig. 2c). The addition of a third unpolarized and unmeasured spin to each qubit enables individual qubit control with only exchange, as shown in Kempe et al.[23], DiVincenzo et al.[24] and Andrews et al.[35], and elaborated in Methods. Unlike most other qubit systems[15,39], no additional physical mechanism is needed to generate entanglement between these encoded qubits. Sequences of exchange-based partial swaps must, however, be designed both to navigate leakage spaces and to be agnostic to the unknown relative states of the unpolarized spins (known as the gauge freedom), as derived by Fong and Wandzura (FW)[28,51] and further elaborated in Methods and Extended Data Figs. 5 and 6.

Qubit preparation and measurement is achieved with Pauli spin blockade. This is a common spin qubit technique that maps spin parity (symmetric versus antisymmetric configurations) to the more easily detected charge configuration[22]. We initialize the outermost dot pairs (P1/P2 and P5/P6), with the X3 gate connecting the innermost, uninitialized electron spins in dots P3 and P4. In Si/SiGe, the fidelity of these state preparation and measurement (SPAM) operations are, in part, limited by the valley excited-state energies of the dots, which we measure using photon-assisted tunnelling to be 70 µeV for dot P1 and 14 µeV for dot P6. The relatively poor valley splitting on P6 is consistent with observations of reduced SPAM fidelity when using sensor M2 to measure Pauli blockade on dots P5/P6, and is why we later prioritize P1/P2 measurement using sensor M1 when possible (Methods). M1-based measurement shows more than 96.0% ± 0.1% fidelity[46] in single-qubit characterizations (Extended Data Fig. 2). Valley splitting may be more reliably large in narrower wells[48], or possibly using other techniques;[52] SPAM fidelity was recently validated to reach more than 99.7% in a similar, narrower-well device[46].

We perform exchange-based control of our encoded qubits using interleaved sequences of baseband voltage pulses that activate nearest-neighbour interactions for a fixed duration $t_{pulse}$, with pulse-to-pulse spacing $t_{idle}$. A single exchange pulse is actuated by voltage-modulating gate X$n$ by approximately 100 mV (and neighbouring P$n$ and P($n + 1$) gates by about −20 mV for capacitive compensation to achieve symmetric operation)[33]. This drives a partial swap $U_{n,n+1}(\theta)$ by temporarily raising exchange energy $J_{Xn}$ from a negligible value to $J_{Xn}/h \approx 100$ MHz (Fig. 2c). The specific angles $\theta$ we used for two-qubit gates are tabulated in Extended Data Table 1 and for single-qubit Clifford gates in Extended Data Table 2, although $\theta$ may be continuously tuned for arbitrary control. We provide control performance metrics in the Extended Data using methods described in Reed et al.[33] and Andrews et al.[35], including an evaluation of random magnetic field gradients due to residual nuclear spins in Extended Data Fig. 1a and control-signal crosstalk in Extended Data Fig. 2b. These metrics are explained further in Methods.

With encoding, SPAM and control in hand, we demonstrate two families of encoded two-qubit gates: CNOT (completing the universal

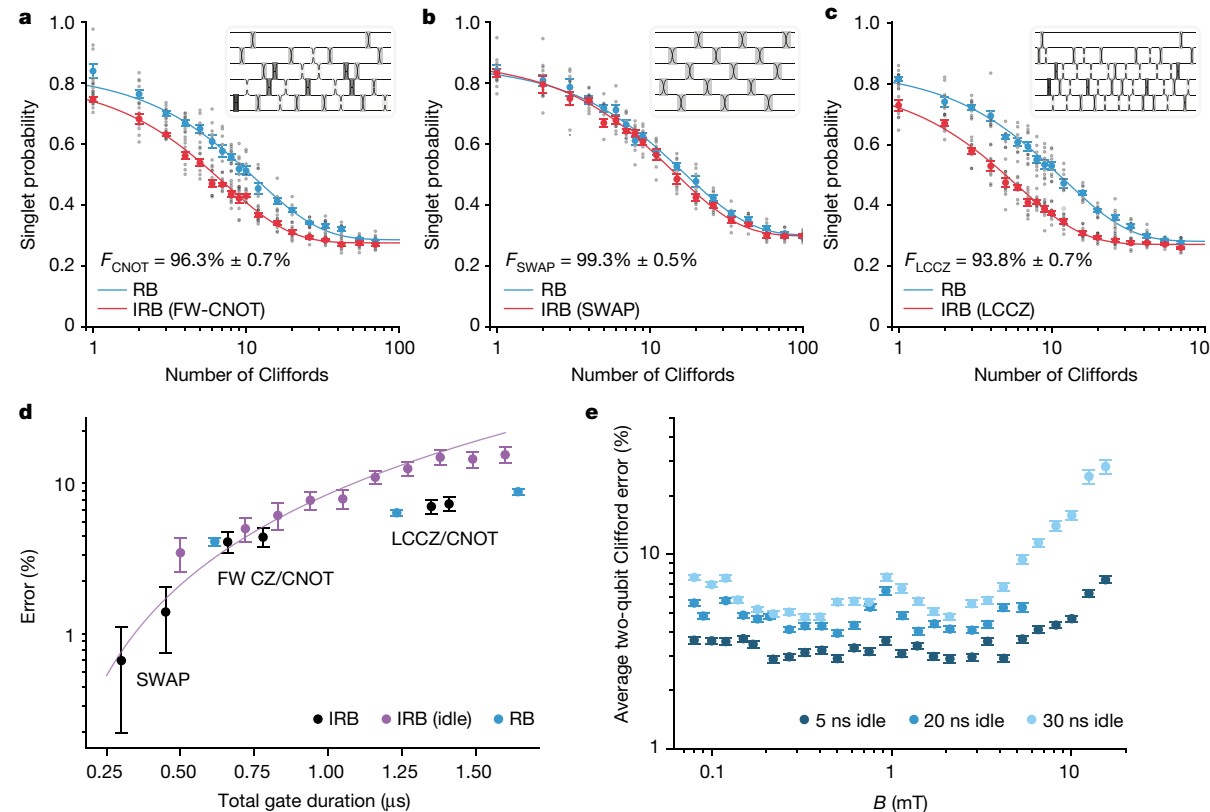

**Fig. 4 | Interleaved two-qubit benchmarking and error trends.**
**a**–**c**, Benchmarking of FW-CNOT (**a**), SWAP (**b**) and LCCZ (**c**) at $B = 0$ mT, with inset exchange pulse diagrams. Here $t_{idle} = 20$ ns for FW-CNOT and LCCZ, but $t_{idle} = 10$ ns for SWAP; $t_{pulse} = 10$ ns for all. Note, the number of Cliffords excludes the final inverting Clifford. **d**, Error as a function of gate duration for selected gates (black) (Methods), for various interleaved idle periods (purple) and for two-qubit Cliffords (blue). Idle error increases with gate duration, closely following a theoretical estimate given by $(t_{idle}/T_2^*)^2$ (purple curve). Some gates show a significant deviation below this curve, indicating that they have some built-in magnetic noise insensitivity. **e**, Average two-qubit Clifford gate error as a function of $B$ and pulse idle time. $B$ is oriented in-plane and perpendicular to the dot array. We see consistent improvement for lower $t_{idle}$, eventually limited by the available bandwidth of the signal chain (not shown). As $B$ increases, we first observe an improvement in fidelity above 200 μT, consistent with the suppression of transverse hyperfine magnetic gradients. The fidelity decreases for $B > 3$ mT because of induced paramagnetic gradients[32], see Extended Data Fig. 4. All dots in (**a**–**c**) represent the average from 1,000 shots of a single random Clifford sequence. All error bars in this figure correspond to 1σ standard deviation intervals.

gate set for quantum computing shown in Fig. 1) and SWAP. The latter is composed of 15 full spin swaps, and is the key resource for moving information through a larger array of dots. We diagram the FW-CNOT and SWAP pulse sequences, decomposed as partial swaps, in Fig. 3a,b. The corresponding gate-specific voltages used for the FW-CNOT case are elaborated in Extended Data Fig. 3. As initial confirmation that we are performing the intended logical operations, we characterize the gates using quantum process tomography (QPT)[53], shown in Fig. 3c,d. We see qualitative agreement with expected tomograms, but, because of the weakness of our QPT inversion method to leakage and SPAM errors, and in particular the poor M2 SPAM in the present device, it is difficult to extract meaningful quantitative results from this protocol. This could, in principle, be mitigated with increased averaging and self-consistent gate set tomography[5,16,54], but we do not explore that here.

Randomized benchmarking (RB) is our preferred method of characterizing gate performance as it is fast, simple, relatively insensitive to SPAM error and requires measurement of only one qubit[55,56]. In RB, a randomly selected sequence of gates that compile to the identity are chosen from a discrete group of qubit operations, typically the Clifford group. This choice depolarizes noise in the encoded subspace, allowing gate performance to be inferred by sweeping the sequence length and fitting return probability to an exponential decay. We generate two-qubit Cliffords by means of standard compilation rules[57]

using the FW-CNOT entangling gate, the aforementioned SWAP gate and single-qubit Cliffords. Of the 11,520 two-qubit Clifford gates, 90% include a CNOT in our composition, 50% include the encoded SWAP and each has an average of 3.1 single-qubit Clifford gates; each operation thus contains 41.1 exchange pulses on average. As shown in Fig. 3e, we find an average two-qubit Clifford fidelity of 97.1% ± 0.2%, far better than suggested by the SPAM-afflicted QPT and comparable to contemporary single silicon spin qubit two-qubit RB errors[7,17], despite using a much larger number of exchange operations per Clifford. (This definition of fidelity incorporates errors due to leakage outside the encoded space but does not separate such leakage from encoded error as in the blind RB protocol[35], for which a two-qubit extension is an ongoing effort, see Methods and Extended Data Fig. 7.) The leakage calculation predicting the observed asymptote of 17/60 ≈ 0.283 is discussed in Methods.

We also measure the performance of individual gates of interest using interleaved randomized benchmarking (IRB)[30]. In this protocol, the operation in question is repeatedly interleaved between a random Clifford gate and the resulting decay is compared with a reference RB decay to infer the operation's fidelity. We perform IRB for FW-CNOT and encoded SWAP in Fig. 4a,b, finding a fidelity of 96.3% ± 0.7% and 99.3% ± 0.5%, respectively. The SWAP error is more than five times lower than that of CNOT, constituting a high-fidelity mechanism for moving data in constrained geometries using this technology. (Spin

shuttling through empty dots[17,18,50,58] is a notable alternative.) A third gate we examine resolves a new potential obstacle to FT posed by our encoding, namely, that gate error can now lead to qubit leakage and the FW-CNOT can spread that leakage throughout a register. To avoid this, a variant of FW-CNOT called the leakage-controlled controlled-$Z$ (LCCZ) was designed to reduce the probability of leakage spreading to approximately the same probability as direct leakage. We find the LCCZ fidelity to be 93.8% ± 0.7% in this device; see Methods.

The dominant error in our EO operation is due to residual hyperfine interactions from the 800 ppm of $^{29}$Si nuclear spins and natural abundance of barrier $^{73}$Ge spins. Evidence for this is provided by the scaling of the sequence error with duration (Fig. 4d), magnetic field dependence (Fig. 4e) and comparison to detailed quantitative numeric simulations, all elaborated in Methods. This varies from studies on single-spin qubits using micromagnetic field gradients, in which isotopic enhancement typically reduces magnetic noise beneath a floor set by charge noise[19–21]. EO is different, as it depends more critically on field uniformity, but is more resilient to charge noise, which we estimate as providing less than 6% of the observed error.

We have demonstrated encoded universal quantum logic operations, including single-qubit gates, CNOT, controlled-$Z$ (CZ), LCCZ and SWAP, using only the exchange-based partial swap interaction and implemented in an extensible physical architecture. This control scheme is distinct from the traditional non-energy-conserving methods of other quantum technologies and may by itself offer near-term opportunities for analogue simulation[59,60]. In the longer term, although the EO DFS encoding offers fundamental advantages, achieving FT with it would require, in part, improved gate fidelity. Given the hyperfine contribution to our error, the primary pathway is to pursue faster control or improved isotopic purification. These refinements could be made independent of SPAM, fabrication yield and crosstalk performance, and are compatible with scaling to larger devices using several back-end metal layers within the SLEDGE platform. This contrasts with lithographically defined microwave-actuated qubits, for which crosstalk reduction and site selectability typically require substantial device redesign with scale. An open question is whether our demonstrated coherent EO universal gate set may leverage states beyond our demonstrated encoding to enable other computational modalities, such as high-efficiency reversible classical computation[45] or permutational QC[61].

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

# Methods

## Encoded subspace notation

In the following sections, we elaborate sources of noise, the theoretical construction of two-qubit gates and further two-qubit benchmarking results. For these elaborations, a more detailed description of the encoded subspace is required, which is best described according to angular momentum quantum numbers. A single spin, say spin 1, has $S_1 = 1/2$ and projection $m_1 = \pm 1/2$. For two spins, labelled 1 and 2, the total angular momentum $S_{12}$ can be $S_{12} = 0$, the singlet state for which $m_{12} = 0$, or $S_{12} = 1$, the triplet of states for which $m_{12} = -1$, 0 or 1. We notate these four states as $|S_{12}; m_{12}\rangle$. The smallest universally controllable EO qubit requires adding a third spin, for which the algebra of angular momentum provides the quantum number $S_{123}$, which can be 1/2 either by adding the single $S_3 = 1/2$ spin to the $S_{12} = 0$ singlet or by adding $S_3$ to the $S_{12} = 1$ triplet. These two choices for $S_{123} = 1/2$ comprise our encoded qubit, and the quadruplet of states with $S_{123} = 3/2$, which also results from adding $S_3$ to the $S_{12} = 1$ triplet, comprise our leaked states. We also have a total spin projection $m_{123} = -S_{123}, -S_{123} + 1, ..., S_{123}$. We thus notate these states here as $|S_{12}, S_{123}; m_{123}\rangle$ and our qubit states are $|0, 1/2; m_{123}\rangle$ and $|1, 1/2; m_{123}\rangle$. The $m_{123}$ degree of freedom is called the gauge, and it is unaffected by exchange operation between any of the three spins. The Pauli spin blockade process on dots 1 and 2 provides initialization and measurement of only the $S_{12}$ quantum number and has no impact on $m_{123}$ (ref. [46]). Singlet measurements on spins 1 and 2 therefore measure the probability of $|0, 1/2; m_{123}\rangle$ independent of $m_{123}$, and provide no information to distinguish encoded triplet $|1, 1/2; m_{123}\rangle$ states from leaked states $|1, 3/2; m_{123}\rangle$. More information about exchange-based gates in a single-qubit system is available in Andrews et al.[35].

## Physical noise sources

We now discuss our techniques for characterizing two well-understood sources of noise and decoherence in our device, that is, magnetic noise and charge noise, and our techniques for limiting crosstalk error. This constitutes a more detailed discussion of Extended Data Figs. 1 and 2 than the main text could contain.

**Magnetic or hyperfine noise.** The DFS encoding we use is predicated on a homogeneous (although possibly time-varying) magnetic field from one dot to the next. Uniform magnetic fields are thus effectively ignored by the DFS encoding, whereas magnetic gradients drive relative precession of electron spin pairs, causing decoherence and leakage out of the encoded subspace. As such, magnetic gradients between spins cause error, both within the encoded subsystem and outside it (that is, leakage.) Magnetic gradient fluctuations, caused by the noisy magnetization of $^{29}$Si and $^{73}$Ge nuclear spins coupled to the electron spins by means of the contact hyperfine interaction, provide the largest contribution to our gate error[32].

We characterize the effective decoherence rate due to hyperfine noise, $T_2^*$, by measuring the decay of a singlet prepared between two electrons and left in the low-exchange 'idle' configuration for a varying amount of time. The ensemble-averaged measurement projection decays to a value associated with the predicted mixture of encoded and non-encoded states: with probability 1/2 to the initial singlet $|0, 1/2; m_{123}\rangle$, 1/6 to the encoded triplet $|1, 1/2; m_{123}\rangle$ and 1/3 to the leaked state $|1, 3/2; m_{123}\rangle$[62]. Spin singlet pairs are initialized on either side of the device then shuttled to and from the desired location by consecutive π exchange pulses. Owing to the relatively poor SPAM fidelity on the M2 side of the device, spin pairs measured on the M2 side show deviations in the decay asymptote from the predicted value of 1/2. The 1/e point of the Gaussian decay defines $T_2^*$, which we measured to be about 3.5 μs for six different spin pairs, as shown in Extended Data Fig. 1a. This timescale is further characterized in Kerckhoff et al.[32] and is fully expected for the present isotopic content. As expected for the number of hyperfine nuclei contributing to this dephasing, there is relatively little variation between each dot[32]. Other silicon qubits with varying levels of isotopic enhancement have been demonstrated elsewhere; for example Xue et al.[5] used similar 800-ppm Si and natural Ge, but Struck et al.[20] used 60-ppm $^{29}$Si, and key demonstrations of long decoherence times approaching minutes have emerged from echo experiments from donor-bound electrons in 50-ppm $^{29}$Si (ref. [31]). In single-dot experiments with thicker quantum wells and micromagnetic gradients, a key learning[19–21] is that isotopic purification of the silicon is sufficient to cause $T_2^*$ to be dominated by transduced charge noise, whereas in thinner quantum wells without gradients, such as the present device, nuclear hyperfine effects continue to dominate, with a clear pathway for improvement of increased isotopic enrichment[32,36].

Unlike charge noise, magnetic noise acts during both pulsing and idling[62] (for example, regardless of exchange energy), and so dephasing occurs continuously during evolution, leading to a total error scaling as $(t_{gate}/T_2^*)^2$. The prefactor of that scaling depends on how much the sequence permutes spins and decouples magnetic noise[32]. In Fig. 4d, we plot the IRB error of idle operations of increasing duration $t$. The calculated theoretical IRB error is $\varepsilon = (120/129)(t/T_2^*)^2$, in good agreement with the data. (The coefficient 120/129 is calculated by the convolution method presented in Merkel et al.[63].) For non-idle operations, error typically falls beneath the $\varepsilon \approx (t_{gate}/T_2^*)^2$ error level of 'doing nothing', because of the partial homogenization of gradient fields due to driven spin exchange. As an example, the LCCZ operation shown here has roughly half the infidelity of an equal-duration idle. The expected IRB error due to magnetic noise for a particular gate built from multipulse exchange sequences is evaluated by numeric simulation[35,62]. We find simulated magnetic error rates to be consistent with the measured data, although they are field dependent, and agreement is limited to uncertainties in the actual magnetic field witnessed at the quantum dot.

Although the EO encoding is explicitly immune to static and fluctuating global fields, global fields do affect the magnitude of local magnetic field gradients. In Fig. 4e, we plot average two-qubit Clifford fidelity measured using RB as a function of both applied field and $t_{idle}$. Here, the magnetic field is aligned in-plane with the device substrate and roughly perpendicular to the linear dot array. We find that error initially decreases by nearly a factor of two with increasing field strength, which is understood as the suppression of nuclear magnetic fluctuations transverse to the field direction, consistent with our numerical simulations[32].

At fields >3 mT, error increases again because of a combination of spin-orbit effects and Meissner screening effects from superconducting parts of the gate stack. The differences in Larmor frequency between the P1 dot and the P2–P5 dots are measured as in Eng et al.[27] and plotted in Extended Data Fig. 4, suggestive of a very small Meissner gradient of order $g|dB/dx|/h = 0.04$ (kHz/mT)/nm, but certainly also including a contribution from spin-orbit coupling[37,64]. The Meissner screening effect was much stronger in earlier devices using aluminium metal[32], but is much weaker in the present device because of the non-superconducting or weakly superconducting TiN gates used in the SLEDGE process[29]. The amount of spin-orbit (which depends on random Ge atom placement) and Meissner screening (which depends on deposition properties and geometry of TiN gates) vary significantly from dot to dot.

**Charge or exchange noise.** Another contribution to our gate error is charge noise, here manifest as exchange noise induced by fluctuations in the lateral trapping and tunnelling potentials. These fluctuations are due to either noise in the signal chain or defects in the gate stack and have a strong $1/f$ spectrum across many decades[65]. This type of noise induces error only during active exchange pulsing, as nearest-neighbour tunnel coupling is suppressed when the associated spins are idling. This idle error suppression results from the exponential scaling of the spin–spin exchange interaction with applied barrier potentials in general, and in particular from the large on/off

ratios that are available with our tightly confining SLEDGE design[29]. As shown in Extended Data Fig. 1b, we quantify this charge noise contribution to our error budget with an exchange oscillation Q-factor at a given $J$ (typically $J/h \approx 100$ MHz). The product of $J/h$ with the 1/e duration of the Gaussian decay envelope of exchange oscillations gives us a number of oscillations, $N_{osc}$. As a budgeting metric, $N_{osc}$ is a superior parametrization, compared with the decay time, as the impact of charge noise on $J$ scales as $|dJ/dV|$ for gate voltages $V$. If $J$ were exactly exponential with voltage, $N_{osc}$ would be independent of voltage, but it is not perfectly so as exchange is subexponential with X-gate voltage[33]. Similar to our magnetic noise heuristic, the estimated theoretical gate error from charge noise scales quadratically with this decay envelope, $1/N_{osc}^2$.

We repeatedly measure $N_{osc}$ as a function of $J$ along the symmetric operation axis, finding that $N_{osc}$ reaches about 50 at 100 MHz, the frequency of operation for experimental convenience. Notably, and contrary to pure exponential activation, $N_{osc}$ rapidly increases from <100 at $J/h = 1$ GHz to >1600 above $J/h \approx 20$ GHz, where we observe a sharp reduction in $|dJ/dV|$ as the potential barrier flattens (Extended Data Fig. 1c). This is the underlying reason that exchange is a subexponential function of voltage, such that the sensitivity to voltage fluctuations, which scales as $|\partial J/\partial V|$, reduces as $J$ increases. We see in Extended Data Fig. 1 that, as exchange asymptotically approaches the approximate double-dot orbital energy, $|\partial J/\partial V|$ nearly vanishes, leading to a marked insensitivity to noise and thousands of exchange oscillations. This phenomenon has been previously reported in GaAs in a different operating regime[66]. The accessibility of this operation point in a SiGe accumulation mode device can be attributed to the large gate action of the SLEDGE design. To resolve coherent oscillations, which occur at exchange frequencies well above the 200-MHz Nyquist frequency of our arbitrary wave generators, we shift the arbitrary wave generator time basis, which is normally set to 2.5 ns, to within the range 2.5–5 ns. A continuous shift of the time basis within this range provides the smooth sampling needed to resolve coherent oscillations without aliasing, and thus to extract both the exchange rate, $J/h$, and $N_{osc}$. Although these exchange energies are too high for practical pulsed operation, they may prove valuable for microwave-sensitive EO encodings[67].

We perform an initial validation of our error model with single-qubit RB in Extended Data Fig. 2a. To properly account for leakage out of our encoded space, we use the 'blind' benchmarking technique described in Andrews et al.[35], whereby sequences are chosen to compile either to the identity or to one of the Pauli gates, and we analyse differences between the $I$, $Z$ and $X$, $Y$ branches. We find an average single-qubit Clifford error of $(1.1 \pm 0.1) \times 10^{-3}$ with a leakage error of $(3 \pm 1) \times 10^{-4}$. This is in approximate agreement with our simulated prediction of error from magnetic and charge noise alone, $5.0 \times 10^{-4}$, but these simulations do not include the effects of pulse distortion or other physical contributions such as Meissner screening or spin-orbit interactions. Evaluation of the impact of charge noise on two-qubit sequences is again done by numeric simulation, and it is found that, for these longer sequences, charge noise provides a smaller contribution to the error, <6%, as indicated in the main text.

**Crosstalk.** A low level of crosstalk is a key ingredient for practical scalability of a quantum register. Although the exchange interaction is intrinsically strongly limited to only nearest-neighbour action[22], thus limiting 'accidental' exchange for pairs not actively pulsed, control-signal crosstalk could still occur if attempting several exchange operations simultaneously in too small a region. The ultimate reason for this is the classical cross-capacitance of metal gates, which leads to a spurious signal in a neighbouring gate, an interaction which falls off as approximately $1/r^3$ for gates separated by physical distance $r$. The effects of this are mitigated in our system in three ways. First, because exchange is an approximately exponential function of its control voltage, a small spurious voltage signal would cause only an exponentially

small deviation of exchange rate at 'idle'. This is because when spins are in the 'idle' configuration, exchange rates are very low ($J_{xn}/h < 10$ kHz) and their derivatives relative to the small voltages that may be generated by cross-capacitance to actively controlled gates are also very low ($dJ/dV/h <$ kHz/mV).

By contrast, when activated by d.c. voltage pulses of order 100 mV, both exchange rates are fast ($J_{xn}/h \approx$ MHz to GHz) and derivatives large ($dJ/dV \approx J/(10$ mV$)$), meaning that simultaneous operation of an exchange pulse for two nearby pairs in an array is best avoided. For the size of array used here, all spins are considered to be 'nearby' for the purposes of crosstalk. This is the reason for our second crosstalk mitigation, which is to ensure that only one spin pair is active at a time, and all others are idle. Future, larger arrays will rely on distant crosstalk signals being small enough to not cause unacceptable errors even with the large values of $dJ/dV$ associated with active exchange, and therefore will allow simultaneous operation. In this case, the approximately $1/r^3$ scaling of voltage signal crosstalk will allow simultaneous operation as long as a zone of exclusion is respected; we expect this zone to be comparable in size to the present device. This constitutes only a relatively small overhead in time, which is well worth the cost of avoiding correlated errors from simultaneous control noise or the need for contextual calibration. We note that both of these 'idle' and 'exchange' regimes, associated respectively with our first and second crosstalk mitigation strategy, are a consequence of the high on/off ratio for exchange evident in Extended Data Fig. 1c and in Reed et al.[33].

Third, voltage pulsing is done using 'virtual gates'[17,59] that compensate proximal barriers to keep nearest-neighbour electrons in idle and align the potential biases to symmetrize the exchange interaction to reduce the sensitivity of the system to charge noise[33].

To demonstrate this low susceptibility to crosstalk when operated with these mitigations in place, in Extended Data Fig. 2b we compare the IRB error of an idle period interleaved between Clifford gates with that of a control operation of equivalent duration performed on the neighbouring qubit. (For the reason explained in the discussion of the second mitigation strategy described above, we emphasize that exchange pulses are never simultaneously applied to generate this or any other data in this experiment.) As expected, because of the low voltage-crosstalk susceptibility at idle, the avoidance of simultaneous pulses and the use of symmetric operation (respectively mitigations one to three), there is no significant difference in error for the two cases to within the uncertainty of the RB measurement. This lack of degradation in performance from operations performed on a neighbouring qubit confirms that our control approach, whereby exchange pulses are applied sequentially to the device without temporal overlap, enjoys minimal crosstalk error.

## Theory of multi-qubit exchange-only pulse sequences

In this section we explain the structure of the entangling pulse sequences we demonstrated, including the purpose and construction of the LCCZ gate.

Unlike the single-encoded-qubit case, when considering operations on two encoded qubits on dots 1–6, the gauges of the two qubits, $m_{123}$ and $m_{456}$, become important. Notating the total angular momentum of all six spins as $S$ rather than $S_{123456}$ and their projection as $m$ instead of $m_{123456}$ for brevity, the two $S_{123} = S_{456} = 1/2$ qubits may combine into an $S = 0$ subspace and an $S = 1$ subspace. Although exchange conserves $m$, still respecting gauge freedom, its action for interqubit operations does depend on $S$, which, in turn, is set by the relative value and phase of $m_{123}$ and $m_{456}$. The gauge freedom, which can be safely ignored for single-qubit gates, must therefore be carefully considered with two-qubit operations.

In 2000, DiVincenzo et al.[24] provided a 19-pulse entangling gate sequence between two encoded qubits that required the total angular momentum of all six spins to be $S = 1$, which was imagined to be accomplished by polarizing the gauge of each qubit. However, the

small spin-orbit effect and long spin relaxation times in silicon leave few good hardware choices for such polarization in our system, limiting the practicality of this approach. A decade later, the FW construction[28] was discovered by computational search using a genetic algorithm. This construction is a gauge-independent CNOT sequence, meaning it correctly performs the same CNOT sequence on the $S_{12}$ and $S_{45}$ degrees of freedom for both $S = 0$ and $S = 1$. Gauge independence allows each triple-dot EO qubit to be initialized as a pair of spin-singlet states and an unpolarized spin, as we do in the present demonstration.

In 2016, Zeuch and Bonesteel[51] showed that the FW sequence is in fact composed of three repetitions of a shorter primitive composite sequence acting on four spins, shown in Extended Data Fig. 5. This primitive sequence is a quasi-Fredkin (controlled-SWAP)[45] gate, swapping the gauge $m_{123}$ with $m_4$ only if $S_{12} = 1$, but not if $S_{12} = 0$. Applying this quasi-Fredkin gate to one qubit on spins 1, 2 and 3 and alternatingly on spins 4, 5 and then 4 again, a $S_{12} = 0$ condition will apply identity three times, whereas an $S_{12} = 1$ condition will swap spins 4 and 5, leaving $m_{123}$ in its initial state (regardless of what that initial state is). These three uses provide a Fredkin gate with a three-spin EO qubit as control and two spins as target. If those two spins are the singlet–triplet pair of an EO qubit, this controlled-SWAP becomes an encoded CZ, the Fong–Wandzura controlled-$Z$ (FWCZ), with no gauge dependence.

Compiling the primitive sequences together (for example, shuffling and combining commuting pulses, removing 2π pulses), one arrives at an entangling gate using 12 π/2 exchange pulses (that is, spin √SWAP gates) on just five fully connected spins. Assuring a CZ adds two more π/2 pulses, and adapting to the linear, nearest-neighbour coupled layout we use here with the measured singlet–triplet pairs on the ends of the array, the FWCZ ends up using all six spins with a further 12 π pulses (swaps on spins) for a total of 26 pulses. The sequence is shown in Extended Data Table 1. We may convert the CZ into CNOT by means of the construction shown in Extended Data Fig. 6a, adding two more pulses. To highlight the physical implementation of such sequences, we present an illustrative example in Extended Data Fig. 3 of the 28-pulse FW-CNOT sequence translated into experimentally accurate voltage waveforms required for device control. The sequences are compiled sequentially, so that no two pulses occur simultaneously, to avoid crosstalk (see Methods above).

The quasi-Fredkin gate on four spins discussed above has an undesired feature, however: if the EO qubit is in its $S_{123} = 3/2$ leaked state, then the gate applies a phase flip to the $S_{1234} = 2$ states relative to the $S_{1234} = 1$ states. Mathematically, no EO four-spin sequence can avoid this problem. As a result, when $S_{123} = 3/2$ and we apply this primitive operation as described for the FWCZ, then when $S_{12345} = 1/2$ and $S_{45} = 1$, the resulting unitary provides a phase flip, and when $S_{12345} = 3/2$, the resulting unitary applies a π rotation about an axis tipped an angle $\tan^{-1}[3\sqrt{(15/11)}\,]$ from the Bloch-sphere $z$-axis to the singlet–triplet qubit defined by $S_{45}$. These operations in general will leak the EO qubit including spins 4 and 5; even when applying the FW gate perfectly, leakage will have spread from one qubit to the next. (Notably, leakage spreads only from the $|S_{12}, S_{123}; m_{123}\rangle$ qubit to the $|S_{45}, S_{456}; m_{456}\rangle$ qubit; if the $|S_{45}, S_{456}; m_{456}\rangle$ is leaked into $|1, 3/2; m_{456}\rangle$, then the gate behaves as intended for $S_{45} = 1$ regardless of $S_{456}$.) This leakage spreading could be highly detrimental to FT, as we are unable to detect leakage directly in general and most quantum error correcting codes are ill-equipped to correct leakage even when it is detected.

The goal of the LCCZ gate is to avoid this leakage spreading when applying encoded CZ gates. The key insights to the LCCZ gate are that the unwanted leakage-induced phase flip or π rotation on the $S_{45}$ qubit are both square roots of an identity operation and all single-qubit operations are identity on leakage spaces. Therefore, if we apply two FWCZ sequences with a single-qubit gate on the qubit composed of dots 1–3 in between, then we will have achieved some controlled-π-rotation gate on the encoded subspaces. The π-rotation angle depends on the choice of single-qubit operation and can be converted

back to Pauli operators $Z$ or $X$ with single-qubit corrections. In particular, we use the construction shown in Extended Data Fig. 6b. The operator $\sqrt{Z}$ (called $S^\dagger$ in Extended Data Table 2, but not to be confused here with total spin) on $|S_{45}; m_{12}\rangle$ is simply another π/2 pulse on these spins, and the single-qubit rotations $R$, $R^\dagger$ and $H$ may be readily derived as exchange sequences similar to those in Andrews et al.[35]. If the CZ in this construction is the FWCZ and the $S_{12}$ qubit is leaky (that is, $S_{123} = 3/2$), then all single-qubit gates have no action and the two FWCZ gates combine into identity, leaving behind only the correctable single-qubit-encoded $\sqrt{Z}$ on the $S_{45}$ pair (which is also $\sqrt{Z}$ on a $|S_{45}, S_{456}; m_{456}\rangle$ qubit). After some compiling, the resulting sequence is 46 pulses for a leakage-controlled CNOT and 44 pulses for an LCCZ; the compiled sequences are also shown in Extended Data Table 1.

The final sequence we demonstrate is encoded SWAP, which is not entangling, but is nonetheless a critical two-qubit gate for moving data and for RB. In many qubit modalities, the SWAP is more complex than the CNOT, as a typical construction uses three CNOT sequences. However, for EO qubits, SWAP is the one transversal operation of the underlying spins; for example, swapping all the underlying spins also constitutes a SWAP of the encoded qubits. If our spins were fully connected, three π pulses to enact spin swaps would suffice. The 15-pulse sequence shown in Extended Data Table 1 performs the permutations needed to move spin information in linear, nearest-neighbour coupled architecture with the three-spin structure reflected about the centre. The timing of these spin-swap operations is chosen so that each spin spends approximately the same amount of time in each dot. A sequence that equalizes time exactly effectively decouples low-frequency magnetic noise;[34,38] the 15-pulse swap shown only partially completes such a permutative dynamical decoupling operation, but is nevertheless much more resilient to magnetic noise than other operations of similar duration.

## Two-qubit benchmarking with bilateral readout and inversion rotations

Although all the RB results in the main text were generated with one-sided readout only, we present here benchmarking results with readout from both sides. As stated in the text, small valley splitting on dot P6 limits the SPAM fidelity and results in a meaningful reduction in measurement visibility. Nonetheless, as Extended Data Fig. 7 shows, the decay rates measured on either side are consistent; we observe similar gate performance using measurement on either side of the device.

In addition, we have explored the RB sequences with and without $X$-gate pre-rotations, reminiscent of the single-qubit blind RB protocol[35] and of character benchmarking on the larger space of encoded qubits[68]. Unlike the single-qubit case, in which linear combinations of the two measurement results can be used to differentiate error in the qubit computational space from leakage error, the two-qubit case navigates through a larger Hilbert space, which requires a greater number of measurements to distinguish those types of error in a similar manner. Although we have not yet formalized such a technique, we still emphasize that the decay rates on the singlet and triplet branches are consistent within the confidence intervals. We take this as a strong sign that the single exponential decay model for each channel is valid here, lending the possibility that linear combinations of exponential decays will enable more detailed leakage analysis in future efforts.

Finally, we explain the different asymptotic singlet probability values at large Clifford numbers for the different RB experiments we present. These may be understood by a relatively simple counting argument, which uses the fact that the total electron spin projection, $m$, is partially conserved in our experiments. This conservation is stronger at higher magnetic fields, where the mismatch of electron and nuclear Larmor frequencies suppress electron-nuclear flip flops and pulse sequences are insufficiently fast to drive the spin system to compensate. Although this conservation law is weaker at low magnetic fields, it is strong enough even in Earth's field that the following counting

argument applies. For one-qubit RB, there are three states for three spins at constant value of $m_{123} = \pm 1/2$; in the $|S_{12}, S_{123}; m_{123}\rangle$ notation: $|0, 1/2; m_{123}\rangle$, $|1, 1/2; m_{123}\rangle$ and $|1, 3/2; m_{123}\rangle$. Magnetic noise will, under application of a large number of Clifford sequences, scramble these and result in a probability of 1/3 of measuring $S_{12} = 0$ at sequence end. For two-qubit RB, $m$ is the projection across all six spins, and half of experiments begin with $m = 0$, a quarter begin with $m = 1$ and another quarter begin with $m = -1$, depending on the random gauge of the two initialized qubits. There are $6!/(3!3!) = 20$ ways to obtain $m = 0$ and $6!/(4!2!) = 15$ ways to obtain $m = \pm 1$. Six of the 20 $m = 0$ states have $S_{12} = 0$ and four of the 15 $m = 1$ states have $S_{12} = 0$, and so if these are again fully scrambled, the expected asymptote is $((6/20) + (4/15))/2 = 17/60 \approx 0.283$. (If we were to instead construct our Clifford gate set using the LCCZ gate, state mixing under $m$-conservation would be less complete and the theoretical asymptote would again become 1/3.) The asymptotic value observed when using the M2 readout is significantly higher than 1/3, as seen in Extended Data Fig. 7b, and is another indication of reduced SPAM performance on this side of the device.

## Data availability

All data generated or analysed during this study are included in this published article and in its extended data.

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

**Acknowledgements** We thank J. B. Carpenter for assistance with all figures, and acknowledge significant technical contributions from E. Acuna, E. Chen, L. Edge, M. Gyure, A. Hunter, C. Jones, B. Maune, S. Merkel, I. Milosavljevic, E. Pritchett, R. Ross, A. Schmitz, C. Schnaible, B. Sun, R. Velunta, S. Wandzura and P. Williams.

**Author contributions** M.T.R., T.D.L. and M.G.B. conceptualized and supervised the project. L.E.E., S.D.H., C.A.J., M.J., G.S., J.W., M.T.R. and M.G.B. designed and fabricated the devices. A.J.W., M.D.R., A.M.J., R.W.A., D.B., J.Z.B., K.E., D.R.H. and J.K. developed hardware, performed measurements and analysed data. J.M. and A.S. developed software to support measurements. B.H.F., T.E.K., A.A.K. and T.D.L. developed gate sequences, device models and simulations. A.J.W., M.D.R. and T.D.L. wrote the manuscript, with input and edits from all authors.

**Competing interests** The authors declare no competing interests.

**Additional information**
**Correspondence and requests for materials** should be addressed to Aaron J. Weinstein or Thaddeus D. Ladd.

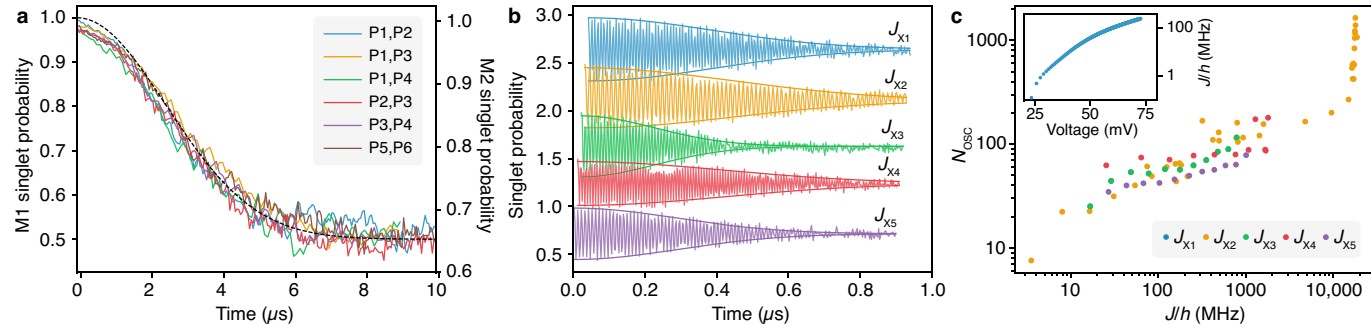

**Extended Data Fig. 1 | Device performance metrics. a**, Magnetic dephasing of a spin singlet at the idle position (where $J \approx 0$), prepared on different dot pairs. The 1/e point of the Gaussian decay envelope defines $t = T_2^*$ and is a simple metric for characterizing the impact of substrate nuclear magnetic noise on qubit performance. We plot a $T_2^* = 3.5\,\mu s$ envelope as a visual guide in dashed black. Most pairs are prepared and measured with high fidelity on the M1 side (left vertical axis); the P5/P6 pair uses M2 (right vertical axis) and has lower contrast. **b**, Charge noise impact on exchange oscillations for each exchange axis. This measurement is analogous to the one for $T_2^*$ except measured at $J/h \approx 100$ MHz such that fluctuations in the exchange energy due to charge noise are the dominant source of decoherence. We parameterize the 1/e Gaussian decay point in terms of the number of coherent oscillations, $N_{osc}$, that occur in that time. Each successive curve is offset on the $y$-axis by 0.5 and on the $x$-axis by 10 ns. Here, $N_{osc} = 57.6, 49.2, 33.4, 70.9$ and $42.3$ at $J/h = 119$ MHz, 84.2 MHz, 114 MHz, 134 MHz and 103 MHz for axes $J_{X1}$–$J_{X5}$, respectively. **c**, $N_{osc}$ as a function of $J$. Owing to the subexponential behaviour of exchange along the symmetric axis[33] (inset), we observe that $N_{osc}$ increases with $J$. At $J/h \geq 17$ GHz, exchange asymptotes as a result of the flattening of the tunnel barrier to a value related to the double-dot orbital energy[66]. In this limit, where voltage excursions on numerous gates exceed ±200 mV, $dJ/dV$ decreases significantly and $N_{osc}$ rapidly increases to an observed maximum >1,600.

**a**

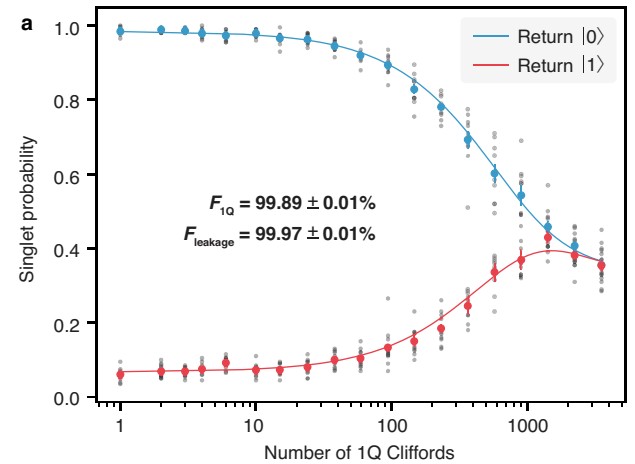

$F_{1Q} = 99.89 \pm 0.01\%$

$F_{leakage} = 99.97 \pm 0.01\%$

**b**

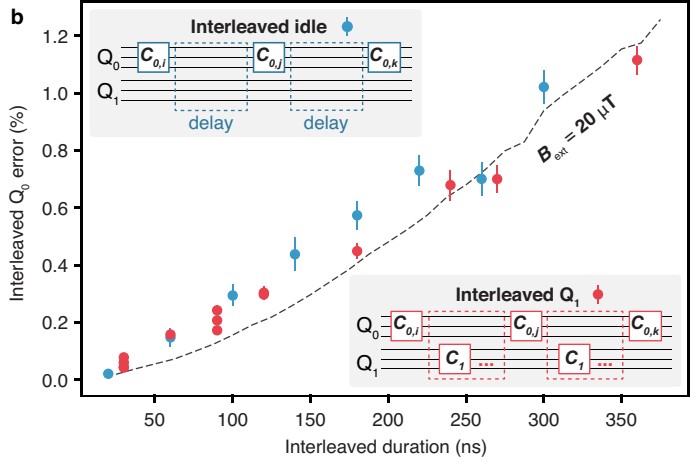

**Extended Data Fig. 2 | Single-qubit performance and inter-qubit crosstalk data. a**, Single-qubit performance for the P1, P2, P3 qubit with M1 readout. The 'blind' RB measurement[35] yields an average single qubit gate error of $(1.1 \pm 0.1) \times 10^{-3}$ and leakage error of $(3 \pm 1) \times 10^{-4}$. Using the methods of Blumoff et al.[46], we infer a SPAM fidelity on the M1 side from these data of $96.0\% \pm 0.1\%$. See Methods for more detailed discussion of this figure. **b**, Crosstalk characterization. IRB error on qubit $Q_0$ (using exchange energies $J_{X1}$ and $J_{X2}$) while interleaving between each Clifford either periods of idle (blue) or composite Clifford gates (orange) on qubit $Q_1$ (using exchange energies $J_{X5}$ and $J_{X4}$). The $Q_1$ composite sequences used here are $X$, $XX$, $XXX$, $Y$, $YY$, $YYY$, $S^\dagger$, $S^\dagger S^\dagger$ and $S^\dagger S^\dagger S^\dagger$. These gates,

respectively, take 90 ns, 180 ns, 270 ns, 120 ns, 240 ns, 360 ns, 30 ns, 60 ns and 90 ns. The $x$-axis is the duration of the interleaved idle period or total duration of the composite pulse sequence on qubit $Q_1$. The black dashed line indicates modelled RB error due to hyperfine dephasing at each idle duration in a magnetic field of 20 µT as generated by a numeric simulation. As discussed in Methods, these data serve as a validation of our approach for limiting crosstalk error, which includes strict avoidance of simultaneous exchange pulses and low susceptibility to spurious signals in the device 'idle' configuration. All error bars in this figure correspond to 1$\sigma$ standard deviation intervals.

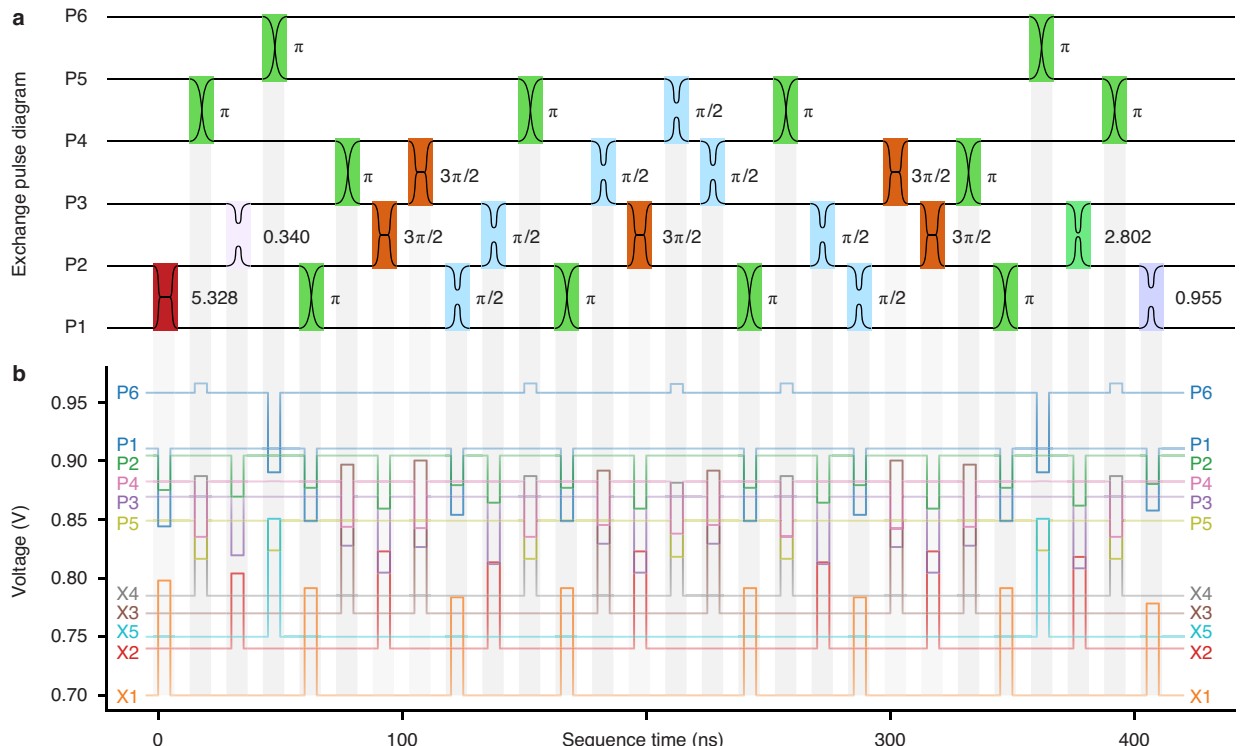

**Extended Data Fig. 3 | A Fong–Wandzura CNOT and its corresponding voltage control waveform. a**,**b**, Pulse angles (**a**) defined on exchange axes are translated into rectangular voltage pulses (**b**) specified at the gates, with varying amplitudes along the corresponding symmetric axes. The voltage waveforms shown here are the exact voltage waveforms used in the RB experiment of Fig. 3c.

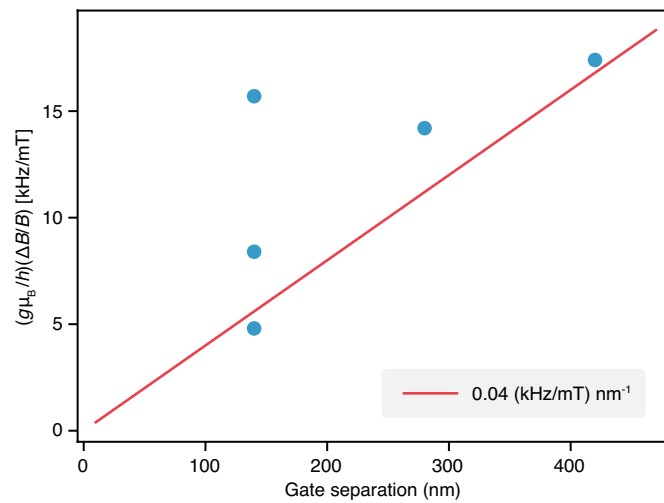

**Extended Data Fig. 4 | Paramagnetic gradients.** Magnetic gradients as a function of gate-to-gate separation in the lateral plane. Gradients are derived from singlet–triplet oscillations of a decay measurement, measured over various spatial separations, as described in Eng et al.[27]. A linear slope of 0.04 (kHz/mT) nm$^{-1}$ is plotted here as a guide to the eye.

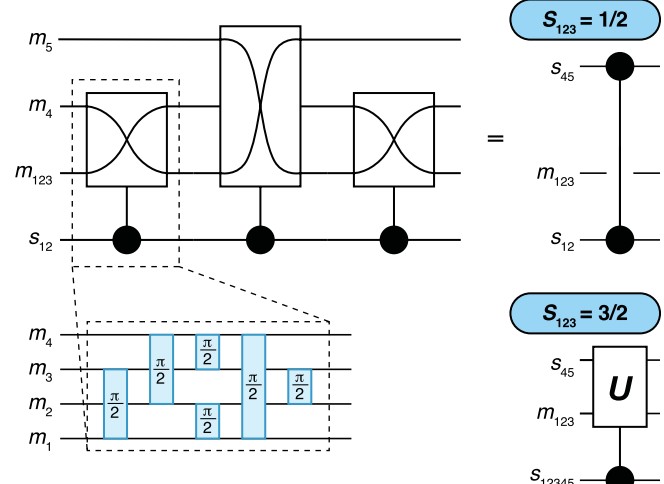

**Extended Data Fig. 5 | Mathematical construction of the FWCZ.** A primitive subsequence coupling all pairs of four spins with $\pi/2$ pulses (spin $\sqrt{\text{SWAP}}$) gates is shown in the dashed box, which, in the $S_{123} = 1/2$ (encoded) subsystem, is controlled by the $S_{12}$ quantum number of the $|S_{12}, S_{123}; m_{123}\rangle$ qubit. The primitive is identity for $S_{12} = 0$ and swaps $m_{123}$ with a fourth spin if $S_{12} = 1$. Three uses with alternating choice of the fourth spin completes a controlled-$Z$ between singlet–triplet subsystems on $S_{12}$ and $S_{45}$. If $S_{123} = 3/2$ (leakage subsystem), the gate applies an $S_{12345}$-dependent unitary $U$ to $S_{45}$, with $U^2 = 1$. A full-pulse construction of this FWCZ including the extra $\pi$ pulses (spin swaps) to adapt to a linear nearest-neighbour layout is shown in Extended Data Table 1.

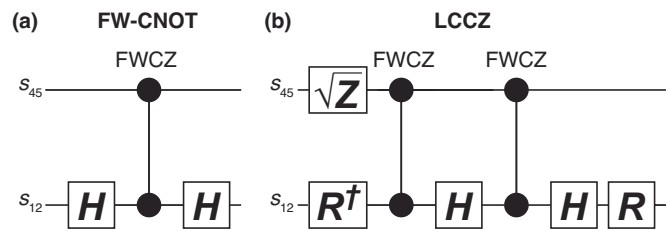

**Extended Data Fig. 6 | LCCZ construction. a**, FW-CNOT is made from FWCZ (Extended Data Fig. 5) in the standard way: two Hadamard gates, notated H, as in Andrews et al.[35] and some compiling. **b**, The LCCZ sequence is made from two FWCZ sequences interspersed with single-qubit gates, and some compiling. Here $R = (I + iZ)/\sqrt{2}$ for identity $I$ and Pauli $Z$. Full-pulse constructions are shown in Extended Data Table 1.

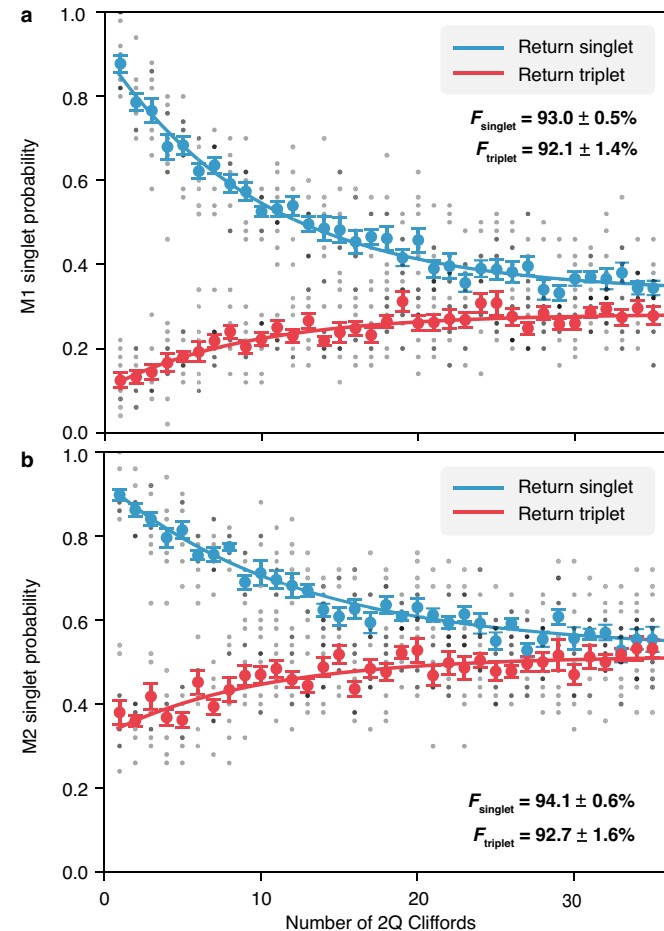

**Extended Data Fig. 7 | Two-qubit RB with two-sided readout.**
**a**,**b**, Benchmarking measurements on the M1 side (**a**) and M2 sides (**b**), with
and without *X*-gate pre-rotations. Cliffords are compiled with the FW-CNOT
entangling gate. We observe worse performance here than for the RB datasets
in the main text as we operate here with $t_{idle} = 20$ ns, compared with $t_{idle} = 5$ ns in
Fig. 3c, but performance here is consistent with that observed in Fig. 4e.

**Extended Data Table 1 | Two-qubit gate sequences**

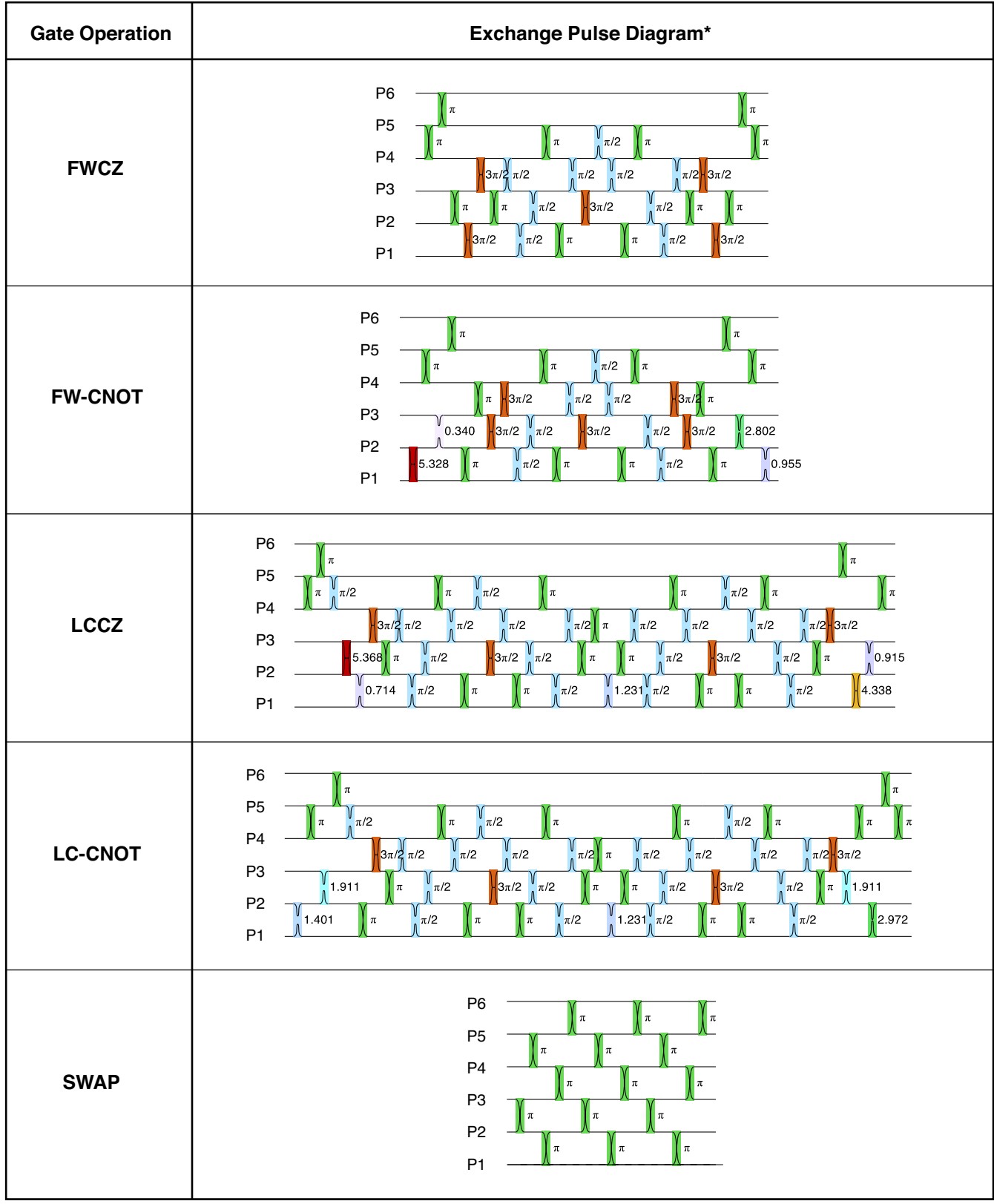

| Gate Operation | Exchange Pulse Diagram* |
|---|---|
| FWCZ | |
| FW-CNOT | |
| LCCZ | |
| LC-CNOT | |
| SWAP | |

*Angles presented here in decimal format are shorthand for the following expressions: $\cos^{-1}(2\sqrt{2}/3) \to 0.340$, $\cos^{-1}[(2-\sqrt{2}+2^{3/4})/3] \to 0.714$, $\cos^{-1}[(-1-2\sqrt{2})/3] \to 0.915$, $\cos^{-1}(1/\sqrt{3}) \to 0.955$, $\cos^{-1}(1/3) \to 1.231$, $\cos^{-1}(1/\sqrt{3}-1/\sqrt{6}) \to 1.401$, $\cos^{-1}(-1/3) \to 1.911$, $\cos^{-1}(-2\sqrt{2}/3) \to 2.802$, $\cos^{-1}[\sqrt{(1/2+\sqrt{2}/3)}] \to 2.972$, $2\pi-\cos^{-1}[(2-\sqrt{2}+2^{3/4})/3] \to 4.338$, $2\pi-\cos^{-1}(1/\sqrt{3}) \to 5.328$ and $2\pi-\cos^{-1}[(-1-2\sqrt{2})/3] \to 5.368$.

**Extended Data Table 2 | Single-qubit Clifford gate sequence**

| Gate | $U_{12}$ | $U_{23}$ | $U_{12}$ | $U_{23}$ |
|---|---|---|---|---|
| $Z$ | $\pi$ | – | – | – |
| $S$ | $3\pi/2$ | – | – | – |
| $S^\dagger$ | $\pi/2$ | – | – | – |
| $H$ | $(\pi - \theta_1)/2$ | $\pi + \theta_1$ | $(\pi - \theta_1)/2$ | – |
| $XH$ | $(\pi + \theta_1)/2$ | $\pi - \theta_1$ | $(3\pi + \theta_1)/2$ | – |
| $YH$ | $(\pi + \theta_1)/2$ | $\pi - \theta_1$ | $(\pi + \theta_1)/2$ | – |
| $ZH$ | $(3\pi + \theta_1)/2$ | $\pi - \theta_1$ | $(\pi + \theta_1)/2$ | – |
| $SH$ | $(\pi - \theta_1)/2$ | $\pi + \theta_1$ | $2\pi - \theta_1/2$ | – |
| $HS$ | $2\pi - \theta_1/2$ | $\pi + \theta_1$ | $(\pi - \theta_1)/2$ | – |
| $S^\dagger H$ | $(3\pi + \theta_1)/2$ | $\pi - \theta_1$ | $\theta_1/2$ | – |
| $HS^\dagger$ | $\theta_1/2$ | $\pi - \theta_1$ | $(3\pi + \theta_1)/2$ | – |
| $HSH$ | $\theta_1/2$ | $\pi - \theta_1$ | $\theta_1/2$ | – |
| $HS^\dagger H$ | $\pi + \theta_1/2$ | $\pi - \theta_1$ | $\pi + \theta_1/2$ | – |
| $S^\dagger HS$ | $\pi + \theta_1/2$ | $\pi - \theta_1$ | $\theta_1/2$ | – |
| $SHS^\dagger$ | $\theta_1/2$ | $\pi - \theta_1$ | $\pi + \theta_1/2$ | – |
| $HSX$ | $\theta_1/2$ | $\pi - \theta_1$ | $(\pi + \theta_1)/2$ | – |
| $S^\dagger XH$ | $(\pi + \theta_1)/2$ | $\pi - \theta_1$ | $\theta_1/2$ | – |
| $HS^\dagger X$ | $2\pi - \theta_1/2$ | $\pi + \theta_1$ | $(3\pi - \theta_1)/2$ | – |
| $SXH$ | $(3\pi - \theta_1)/2$ | $\pi + \theta_1$ | $2\pi - \theta_1/2$ | – |
| $X$ | – | $\pi - \theta_1$ | $\theta_1$ | $\pi - \theta_1$ |
| $SX$ | – | $\theta_3$ | $\theta_2$ | $\theta_4$ |
| $S^\dagger X$ | – | $\theta_4$ | $\theta_2$ | $\theta_3$ |
| $Y$ | $\pi$ | $\pi - \theta_1$ | $\theta_1$ | $\pi - \theta_1$ |

*Angles used here are: $\theta_1 = \tan^{-1}\sqrt{8} \rightarrow 70.529°$, $\theta_2 = \pi - \tan^{-1}(\sqrt{5}/2) \rightarrow 131.81°$, $\theta_3 \rightarrow 74.755°$ and $\theta_4 \rightarrow 201.625°$.