## [Peer Review File · Nature]

Manuscript Title: Universal logic with encoded spin qubits in silicon

Reviewer Comments & Author Rebuttals

Reviewer Reports on the Initial Version:

Referee #1 (Remarks to the Author):

Universal logic with encoded spin qubits in silicon

The manuscript by Weinstein et al. presents a two-qubit quantum processor in silicon capable of performing universal quantum logic, this is two-axis single qubit control and entangling two-qubit gates. The qubit embodiment is the exchange-only qubit which is physically constituted by three spins and the single qubit operations are performed exclusively by utilizing the spin-spin exchange interaction. The resulting architecture then consists of 6 spins, isolated in 6 gated-defined quantum dots in an isotopically purified Si/SiGe quantum well (for enhanced spin coherence) which are distributed in a line. Furthermore, the spins are readout by sensing quantum dots located in close proximity to the chain of quantum dots using spin-to-charge conversion via Pauli spin blockade.

To my knowledge, this is the first demonstration, on any material, of two-qubit operations on the exchange-only spin qubit. This is an important milestone since the exchange-only qubit can be controlled all-electrically by modulating the exchange interaction strength using controlled gate pulses. Electrical as opposed to magnetic control of spin qubits is interesting from the point of view of scaling as it does not require further infrastructure besides the gate electrodes that are used for quantum dot definition and coupling. Further, in this work, authors achieve good two-qubit gate fidelities (as well as single qubit gates) with an average of 97.1%. This required of a substantial number of well-calibrated exchanged pulses involving up to 5 different interactions indicating a high degree of technical prowess.

Although the results are remarkable and deserves wide dissemination in a high impact journal, I have my doubts it guarantees publication in Nature. Let me elaborate my reasons:

1. Recently, three publications have appeared in Nature showing silicon-based quantum processors with exquisite single- and two-qubit gate fidelities, beyond the thresholds required to implement the quantum error correcting surface code: Nature 601, 343–347 (2022), Nature 601, 338–342 (2022), Nature 601, 348–353 (2022). Further in Science Advances another demonstration with >99% control fidelity and >97% initialization and readout fidelity has been published DOI: 10.1126/sciadv.abn5130. The current manuscript does not represent a technical improvement over any of these manuscripts: Average two-qubit gates are below the state-of-the-art and the readout (initialization) problem due to low valley splitting in Si/SiGe does not seem to have been resolved. This is particularly limiting in this work, specially on the M2 sensor side of the structure.
2. In the aforementioned publications in Nature, there are demonstrations of use cases where the processors are used to run quantum algorithms or generate Bell pairs and even GHZ states. The manuscript under review stops at the demonstration of the two-qubit gate implementation.

Overall, as I said, this manuscript is a tour de force of technical prowess and therefore should be considered in high impact journals but in its current form it is not superior to the state-of-the-art. Therefore, I am not convinced Nature is the best host for this work.

Besides, my overall view of the manuscript, I would like to highlight a few aspects that could improve the manuscript.

1. The abstract has references to terms that I feel have not been elaborated sufficiently in the text.
(i) Low control cross talk: the operations are not run in parallel and hence a quantitative assessment of cross talk is lacking. (ii) Thousands of pulses: could the authors be more quantitative?
2. The authors discuss that exchange is exponentially suppressed spatially. Spin-spin exchange interaction drops with $1/r^3$. Could the authors clarify?
3. Could the authors comment on the scalability of the approach? At the moment, the readout is done using the two outer most quantum dots of each trio while the two-qubit interaction is mediated using dots in the inner most locations. Would there be any issues or modifications to the protocols that would need to be introduced to couple a third qubit, for example?
4. Could the authors elaborate why the T_2^* are identical? It is remarkable that there is quasi-negligible spread.
5. Appendix I, explaining the exchange-only qubit and the two-qubit operations is very informative. However, for the description of the single qubit gates one needs to refer to citation [1]. Would it be possible to include a description of the single qubit gates in the appendix, so the article becomes more self-contained?
6. Fig 2. It is stated that to read the P5/P6 pair the M2 sensor is used and has lower contrast. However, I cannot find data with lower contrast in the Fig.
7. Fig 2. Could the authors elaborate on the statement that at $J/h > 17\text{GHz}$ the exchange asymptotes due to the flattening of the tunnel barrier?
8. Fig 3. a,b. There is no scale for the shading of the gate operations, i.e. the angle.
9. Fig 3 caption. Could the authors give more details about the differences between the two devices studied?
10. The valley-splitting for dot 6 is low. Is there a plan to overcome this random variation of the valley splitting? Could readout have been performed on another dot?
11. A statement about future larger devices may support simultaneous operation is made. Could the authors elaborate?
12. Could the authors elaborate on the weakness of the QPT inversion method to leakage?
13. The authors indicate that the high-fidelity SWAP gate in the EO modality may enable new device topologies. Could the authors elaborate?
14. The authors discuss that the TiN gates used in SLEDGE do not superconduct. Could the authors explain why and what the parameters of the film are?

Referee #2 (Remarks to the Author):

The manuscript by Weinstein et al. reports on all-exchange-controlled single- and two-qubit gates in a linear array of six quantum dots, forming two triple-dot qubits. The experiments demonstrated are extremely technically challenging, and it is a feat to reliably perform the voltage pulses and

calibrations required to operate these complex quantum gates in a multi-dimensional gate space. The manuscript is very well written, and the described experiments constitute a step into a promising direction, especially with the streamlined fabrication process developed by the authors in their previous works. I congratulate them on getting all aspect of the difficult experiment to work.

In the end, however, I am not completely convinced, that the experiment at its current stage surpasses the high bar of novelty and demonstrates the technological leap forward justifying a publication in Nature. A few concrete arguments that come to mind are:

The main advance of the manuscript constitutes the ability to perform two qubit gates in an all-electrical system, as opposed to single-spin systems where magnetic fields and micromagnets are required. While this is a very welcome development for the exchange only qubit, two-qubit gates have been implemented in many different platforms, including multiple quantum dot platforms and by many groups in the same SiGe platform, previously. There is therefore limited novelty in the concept itself. Furthermore, the method (gate-exchange) to perform the two-qubit gates does not use a new insight, but is a demonstration of what is known to be possible. In my opinion, the fabrication and technological feat of manufacturing and operating the array is what is the real meat of the paper. However, while still extremely impressive, it makes the manuscript perhaps better suited for a more technical journal.

I also considered the question of whether the paper demonstrated fundamentally new physics, functionality, or ideas from another discipline. Here, I think that two-qubit gates and SWAP operations based on exchange, singlet-triplet readout, in sub-ten quantum dot arrays have been reported before, and the authors have themselves demonstrated the SLEDGE architecture in previous manuscripts.

As for profound impact from a technological or scientific perspective, despite the impressive results, it is difficult for me to see how this architecture would form the building block of future scaled systems, especially as other groups have shown modular designs with a clear view towards going towards 2D. Perhaps the manuscript could be strengthened with a clear statement about scaling of the architecture, or a citation to relevant proposals. It is especially difficult to see how the gate design here may extend to exchange coupling in the second dimension. To be most impactful at this stage of the spin qubit field, a qubit array on a two-dimensional chip should be able to demonstrate, or at least go towards, scalability and two-dimensionality. For these reasons, I am not completely comfortable with recommending publication in Nature.

I had some further comments on some specific questions:

- 1) For few-qubit arrays such as this, crosstalk is an important concern to address for the gate-defined quantum dot community. Is any calibration done to assess the effect of crosstalk (for example, single-qubit benchmarking applied only to the left triple-dot qubit, with a certain exchange pulse fidelity, followed by activating single qubit gates on the second qubit, showing no effect on exchange in the right two dots), or are the pulses and exchange gates tuned up concurrently, assuming the crosstalk remains more or less constant, and it is accounted for in this way? In my opinion, tuning up one set of pulses independently and then expecting the same effect when pulses on the other qubit are added in, is difficult. The authors only refer to this in the second page, stating that exchange gates benefit us by “natively providing low control crosstalk”, without any citation or evidence (while on page 4 they say that the pulses were not applied simultaneously, an important capability for a future quantum processor, precisely to avoid crosstalk).

2) Figure 1 is compelling and well designed. However, the materials making up the various components in the top-view and cross section are not described well either in a legend or in the caption. This should be rectified as I had to dig into the text and other papers to understand the design, especially the patch in the centre of 1(a).

3) Overhead: The number of exchange pulses required to perform a two-qubit gate are many, with probably some probability of error per pulsed operation; this way of encoding qubits therefore seems to come with immense overhead and calibration/operational complexity. The authors suggest in the supplementary (page 15) to use magnetic decoupling pulses and in page 5 of the main text to use gate set tomography to improve their fidelities (by the way, they should cite Kuemmeth, UCPH (Nat. Nano , 2017) and Bluhm, Aachen (Nat. Comm 2017) for decoupling experiments and gate set calibration experiments). These further sequences will add even more complexity to the protocol for even a single two-qubit gate. The number of pulses required means that (Figure S3) a single Fong-Wandzura CNOT (without decoupling pulses) takes about 400ns to perform, while the idle singlet state $T2^*$ is stated as 3.5us. The LC-CNOT is 46 pulses, and the LCCZ is 44 as well. This seems like a pretty bad scaling factor for performing multiple gates before decoherence; can the authors comment on this overhead?

5) Noise analysis

The noise analysis, in my opinion, came across as the weakest point of an otherwise strong manuscript. I was unable to convince myself that magnetic noise is indeed the limiting factor. Figure 4e seemed to me to show that magnetic field does not in fact affect Clifford fidelities much until at fields $>3\text{mT}$ the superconducting gate stack becomes involved. The papers cited in support of the analysis that magnetic field is the main contribution to the error (Ref 1, 22, 34) seem to all be self-cites. More varied citations, perhaps to theory by other groups, or a more clear model or demonstration of the errors from charge noise vs magnetic noise is required, I think, to make the statement the authors make that in their system “The dominant error source of all entangling gates is hyperfine magnetic noise”. Lastly, if the magnetic noise hypothesis were to be true, the authors suggest that the SiGe layers on either side of the quantum well may need to be isotopically purified, which perhaps also impacts scaling? How far from the quantum well does the wafer need to be isotopically purified?

6) Leakage: It is known that the EO qubit gains all-electrical control but makes the sacrifice that it is more vulnerable due to a larger leakage subspace. The authors state that they are not completely sure about the attribution of leakage error versus encoded errors via randomised benchmarking. I think this goes along with point (5) in that it is difficult to parse (after reading the entire paper a few times) what is actually the top limiting factor for the two-qubit fidelity, and whether the authors know this clearly, and how they hope to tackle it next. Is the limitation indeed the magnetic noise and is leakage not a problem? In the outlook they seem to intend to work on it in the future to reach fault tolerance.

6) I also had some comments about referencing: the paper has quite a few self-cites: 9, including authors at the same institution. They should also, or instead, cite other groups on demonstrating spin qubit arrays (especially 2D arrays, such as Germanium in Delft and GaAs in Delft, Copenhagen,

Grenoble), gate set tomography (Aachen), and symmetric operation; this would also bolster confidence in the section on noise analysis (especially in the last two paragraphs before the conclusion).

Referee #3 (Remarks to the Author):

The encoded qubit scheme for quantum dot spin qubits has been proposed and explored for over two decades. It requires at least three physical qubits for one encoded qubit, and the early progress in experimental technology in quantum dot spin qubits has been rather slow, making it a challenge to demonstrate this idea. More recently the progress has been accelerating and multiple quantum dot spin qubit devices have been realized. This work experimentally demonstrates the single- and two-qubit gate operations of the encoded qubit in a linear array of six quantum dots.

Encoded qubit is defined in the decoherence-free subsystem, and gate operations for encoded qubits can be implemented using only the exchange interactions between physical spin qubits. Since exchange interactions can be easily controlled electrically by tuning the gate voltages on the metallic leads, it provides a viable route for scalable quantum computers.

The authors previously demonstrated single-qubit gate operations and now add two-qubit gate operations to complete the universal set of gates. It is an important step for encoded qubit quantum computers, and also shows the capability of current state-of-the-art multiple QD devices. The reported gate fidelities are not yet at the surface code threshold level, but at a high 90's % with some room for improvements. On the other hand, although a number is not given in the manuscript, the SPAM errors still seem to be significant.

Another important advancement is leakage-controlled gate operations. Encoding a qubit state in a larger quantum system introduces the possibility that the quantum system can be outside of the encoded qubit subspace, making it vulnerable to leakage errors. Although the leakage-controlled gates did not lead to higher fidelity (probably because it requires a longer sequence of exchange control pulses), it may be essential for further development of encoded qubit devices.

I think this manuscript is well written and it warrants publication in Nature.

Here are some comments I'd like the authors to address:

1. SPAM error seems to be quite large yet. What are the main causes and what will be a plausible strategy to improve it?
2. It is briefly mentioned that the exchange pulses were applied sequentially to limit crosstalk. I thought one of the main advantages of the exchange-only encoded qubit is that all controls are done by simple voltage controls and the crosstalk is not a significant issue once calibrations are done. If crosstalks limit the parallel application of voltage pulses, then it may be a serious problem for the future development of larger systems. Any comments on this?

3. The exchange control is vulnerable to charge noise and it is beneficial to perform the operations on a sweet spot or symmetric point. In this multiple quantum dot device, how the sweet spots are determined experimentally? Can parallel applications of exchange pulses be done while maintaining the sweet spots, as is expected in theory (Shim and Tahan, PRB 93, 121410(R), 2016)?
4. For two-qubit operations between two exchange only qubits, there is a scheme that requires a much shorter sequence of pulses based on the difference in strengths in intra-qubit and inter-qubit exchange interactions (Doherty and Wardrop, PRL 111, 050503, 2013). Is it plausible in this device?
5. Other than the larger number of physical qubits required, what do you see as the biggest huddle for scaling up these encoded qubit devices?

-- END --

REVIEWER 1

To my knowledge, this is the first demonstration, on any material, of two-qubit operations on the exchange-only spin qubit. This is an important milestone...

We thank the reviewer for his or her time spent with our manuscript, and we are happy to see the importance of this milestone acknowledged.

1. Recently, three publications have appeared in Nature showing silicon-based quantum processors with exquisite single- and two-qubit gate fidelities, beyond the thresholds required to implement the quantum error correcting surface code: Nature 601, 343–347 (2022), Nature 601, 338–342 (2022), Nature 601, 348–353 (2022). Further in Science Advances another demonstration with >99% control fidelity and >97% initialization and readout fidelity has been published DOI: 10.1126/sciadv.abn5130. The current manuscript does not represent a technical improvement over any of these manuscripts: Average two-qubit gates are below the state-of-the-art and the readout (initialization) problem due to low valley splitting in Si/SiGe does not seem to have been resolved. This is particularly limiting in this work, especially on the M2 sensor side of the structure.

If we wish to do direct comparison of fidelities, we should make two considerations. First, a quantum computer is not controlled by one particular gate, but rather by complex sequences of one- and two-qubit gates. In this respect, the most important fidelity is that of a significant random gate sequence (and not one particular two-qubit entangling gate optimized for fidelity), for which a complete two-qubit Clifford gate has emerged as a standardizing example. In this regard, the two-qubit Clifford fidelity of 97.1% we report is, in fact, on par with the only two-qubit Clifford demos mentioned in those Nature papers [1–3] (Noiri et al., at 98.7%; the other two did not perform benchmarking), and superior to the Science Advances demo (Mills et al., 93.1% [4]). This parity is remarkable given that the number of fundamental exchange pulses our demonstration employed averaged to 41.1 per Clifford rather than the two or three used in those other papers, indicating that the fidelity of the fundamental gate set – here, the partial spin-swap – is significantly better in our device, where field gradients are not used. It is a combination of this lack of field gradients (which prevent charge noise from coupling to spins directly) and the quality of our device and tune-up procedure which makes the difference, and therefore a “technical improvement” relative to the papers cited.

Concerning SPAM, one measure for this is the visibility of single-qubit RB. Although one of our sensors was indeed worse than the others due to valley splitting concerns, the M1 sensor reading out dots P1 and P2 was at or in excess of the state-of-the-art for quantum dots. Relative to the quantum dot papers indicated by the reviewer, the Delft result [2] indicated did not perform RB and did not re-

port the SPAM measures from GST, the RIKEN result [1] had a visibility of 60% in single-qubit RB (worse than even our valley-afflicted M2 SPAM), and the Princeton Science Advances paper [4] indicated about 3% error (e.g. 97% contrast). Our former Fig. 2(d) and present Fig. Extended Data 2 shows our single-qubit RB contrast at about 99%. In a similar device, we performed a detailed measure of SPAM and found 99.75% contrast, cited as Blumoff et al., 2022 [5].

The point of our manuscript, however, is not to claim a significant improvement in fidelity or SPAM – as we say, the results are comparable. The important reason for exposure in Nature is because, unlike in any of the works listed, we employ a decoherence-free subsystem encoding and perform universal action within that encoding, a key step on the road to QEC and FT. As we write, this is built on an entirely separate computational model based on SWAP rather than NOT (or single-spin Pauli) operations.

2. In the aforementioned publications in Nature, there are demonstrations of use cases where the processors are used to run quantum algorithms or generate Bell pairs and even GHZ states. The manuscript under review stops at the demonstration of the two-qubit gate implementation.

In our opinion, our demonstration involves much more complicated quantum manipulations than GHZ or Bell state production. The circuits used to produce Bell or GHZ states involve only a few (typically one or two) entangling gates. In randomized benchmarking, by contrast, we run numerous circuits with up to one hundred two-qubit Clifford operations. The sequences must be precisely controlled to properly invert to the identity (e.g. they are not actually “random”). In a sense, randomized benchmarking derives its power from the fact that it is running many, many random algorithms.

We also perform process tomography, the constituent data set of which incorporates numerous highly entangled *encoded spin qubits*. Ignoring the encoded nature of this demonstration, we have generated numerous computational states with Bell-like entanglement as a part of this data acquisition. Viewed from the perspective of single spins, however, our quantum manipulations are creating entangled states of three-spin-entangled basis states. The resulting states analyzed by tomography are therefore very highly entangled states of all six spins, which are much more nonclassical and far larger than a single 3-electron GHZ state.

We do not see what an algorithm demonstration would add to the key performance metrics we have provided. (Only one of the three recent Nature/Science Advances papers indicated ran an algorithm, employing a total of two entangling gates on two spins). There should be no doubt however, given our randomized benchmarking capability and fidelity, that our system is more than capable of executing a less complicated sequence like the ones in the compared papers.

3. The abstract has references to terms that I feel have not been elaborated sufficiently in the text. (i) Low control cross talk: the operations are not run in parallel and hence a quantitative assessment of cross talk is lacking.

The statement referred to the intrinsically low crosstalk of the exchange interac-

tion, which contrasts to other qubit types (superconducting, ion-trap, etc.) which often have higher-crosstalk interactions (microwave RF fields, lasers). However, to better substantiate this claim, we have now added to the Extended Data and to the Supplement an additional data set to show this low crosstalk by interleaving RB on one encoded qubit with quantum action on a second, showing no adverse effect of the parallel operation. As advertised, this is a very strong test validating our implicit claim that controlling one of our qubits has no effect (or rather, the equivalent effect of an idle operation) on the other. This now is briefly referenced to in the main text (paragraph 9: “... and control signal crosstalk ...” and new Fig. Extended Data 2) and discussed at length in Section I.C of the supplement.

3.(ii) Thousands of pulses: could the authors be more quantitative?

Our RB data (Fig. 4) extends to 100 two-qubit Clifford gates, which, as we explain in the text (paragraph 11: “each operation [...] contains 41.1 exchange pulses on average”), means just over 4100 individual exchange pulses.

4. The authors discuss that exchange is exponentially suppressed spatially. Spin-spin exchange interaction drops with $1/r^3$. Could the authors clarify?

We appreciate the question, as this is a key point. We have now elaborated on this statement at length in the Supplement, again section I.C. To briefly reiterate here: it is true that capacitive crosstalk between gates drops roughly as $1/r^3$, and as such if two nearby exchange pulses were on simultaneously, such crosstalk would affect our system. However, as the exchange-only modality is amenable to short “burst” pulses, these can be fired to avoid simultaneity within an appreciable spatial area, and in conjunction with the very high “on/off” ratio of exchange, actual exchange crosstalk may be rendered negligible. The distance at which simultaneous control pulses could safely be applied without meaningful error is somewhat larger than the present device, so we do not employ simultaneous control here.

5. Could the authors comment on the scalability of the approach? At the moment, the readout is done using the two outer most quantum dots of each trio while the two-qubit interaction is mediated using dots in the inner most locations. Would there be any issues or modifications to the protocols that would need to be introduced to couple a third qubit, for example?

This approach is highly amenable to scaling, and a third qubit could be straightforwardly added to the line without any significant change in fabrication modality or operational principles. Our SLEDGE architecture also makes it possible to add whole additional rows of dots, albeit at a cost of increased fabrication complexity (with more back-end-of-line steps). We now include Fig. 2(d) a schematic of a potential 2-d device, to elaborate how additional rows can be added, and we put more emphasis on the fact that our high-fidelity encoded SWAP enables the scaling of this design where measurement only occurs at the edges of the an array (paragraph 6: “... sufficient connectivity is nevertheless available to enable

two-dimensional codes when leveraging the high-fidelity native SWAP operation that we demonstrate below.”).

6. Could the authors elaborate why the T_2^* are identical? It is remarkable that there is quasi-negligible spread.

The amount of spread is expected given the number of nuclear spins contributing to T_2^* , whose origin is entirely hyperfine. A number of N in the hundreds of ^{29}Si and ^{73}Ge nuclei contribute to this dephasing, and the deviation in T_2^* goes as $1/\sqrt{N}$, or about 10%. Such numbers are elaborated in Kerckhoff et al. PRX Quantum 2, 010347 (2021) [6], which we cite amongst our indication of this point now in Supplement section I.A.

7. Appendix I, explaining the exchange-only qubit and the two-qubit operations is very informative. However, for the description of the single qubit gates one needs to refer to citation [1]. Would it be possible to include a description of the single qubit gates in the appendix, so the article becomes more self-contained?

We have copied the table from (former) citation [1] into the Extended Data Table 2 for convenience, alongside the two-qubit sequences previously in the supplement, now including the same symbolic notation for partial swaps.

8. Fig 2. It is stated that to read the P5/P6 pair the M2 sensor is used and has lower contrast. However, I cannot find data with lower contrast in the Fig.

As explained in the caption, the axis on the left is for the T_2^* data exclusive of P5/P6, and the axis on the right is for P5/P6, i.e. data obtained utilizing readout on the M2 side of the device. The right axis has a different scale than the left.

9. Fig 2. Could the authors elaborate on the statement that at $J/h > 17$ GHz the exchange asymptotes due to the flattening of the tunnel barrier?

We now elaborate in section I.B of the supplement. This is effectively the same phenomenon as previously observed in GaAs double dots in a different operating regime at Dial et al. [7].

10. Fig 3. a,b. There is no scale for the shading of the gate operations, i.e. the angle.

Indeed. We feel that indicating these angles makes the figure too busy to be effective. As such, we now provide a reference in the caption to Extended Data Table 1, which includes all angles explicitly enumerated, and so also provides the key to the shading.

11. Fig 3 caption. Could the authors give more details about the differences between the two devices studied?

The second device is a sister to the first, coming from the same wafer and having the same well width, gate pitch, and similar SPAM. It was less extensively studied than the device presented throughout the manuscript. A note of the similarity has been added to the caption (Fig. 3. (c-d): “The CNOT tomogram was measured on a similar device from the same wafer, with the same well width, gate pitch, and comparable SPAM”).

12. The valley-splitting for dot 6 is low. Is there a plan to overcome this random variation of the valley splitting? Could readout have been performed on another dot?

It is true that the excited state energy in dot 6 was relatively low and that without a reliable solution, such issues would pose an obstacle for this technology. As such, there is a significant ongoing research effort into methods to increase valley splitting. This topic is not the focus of the present paper, but we have added citations to some of the latest thinking, Refs. [8 and 9].

Readout was indeed performed on other dots. In our device arrangement, swapping spins across an array is high fidelity, and hence bad readout sensors can be avoided by swapping to good ones. For example the T_2^* measurement in Extended Data Fig. 1 on dots 3 and 4 were shuttled to dots P1/P2 for readout, not P5/P6. However, for the method of measurement we employ (See Blumoff et al. [5] for details) dots used for readout require a tunnel gate connection to a bath (i.e the T1/T6 gates in Fig. 1).

13. A statement about future larger devices may support simultaneous operation is made. Could the authors elaborate?

We discuss this in our answer to question 2. While the present device is not large enough for simultaneous operation (at least not with extra “contextual” calibration), we anticipate devices only marginally larger can start exploring this frontier.

14. Could the authors elaborate on the weakness of the QPT inversion method to leakage?

While we very much appreciate the interest in this topic, rigorous quantum characterization, verification, and validation (QCVV) of multiqubit devices in the presence of leakage is an area of ongoing research in our group and others; the state of the art is too complex to add to this paper or to this response. As we say, part of the reason we prefer to use randomized benchmarking in the present result is that it is better understood at the present moment.

15. The authors indicate that the high-fidelity SWAP gate in the EO modality may enable new device topologies. Could the authors elaborate?

Our discussion is pointing out that the high-fidelity SWAP demonstrated in this work allows for sufficient movement of the qubit states/data so as to enable

two-dimensional codes even when limited to arrays of qubits of finite depth in 1 dimension. We have modified our discussion in the text to more clearly explain this, (paragraph 6: “... sufficient connectivity is nevertheless available to enable two-dimensional codes when leveraging the high-fidelity native SWAP operation that we demonstrate below.”).

16. The authors discuss that the TiN gates used in SLEDGE do not superconduct. Could the authors explain why and what the parameters of the film are?

This is atomic-layer deposited (ALD) TiN of small critical dimension, for which superconductivity is not a guarantee. See, for example, Baturina et al. [10], for a flavor of the complexity of phases of thin ALD TiN nanowires. We in fact cannot know for certain which parts of the gate stack are superconducting; we do see drastically reduced Meissner screening relative to aluminum gates (cf. Kerckhoff et al.[6]), although whether this is because the gates fail to superconduct entirely or whether the field penetration depth is larger than the film thickness is uncertain. Film parameters are available in the cited Ha, et al. paper [11]. We now make this elaboration in section I.A of the Supplement.

I. REVIEWER 2

The manuscript by Weinstein et al. reports on all-exchange-controlled single- and two-qubit gates in a linear array of six quantum dots, forming two triple-dot qubits. The experiments demonstrated are extremely technically challenging, and it is a feat to reliably perform the voltage pulses and calibrations required to operate these complex quantum gates in a multi-dimensional gate space. The manuscript is very well written, and the described experiments constitute a step into a promising direction, especially with the streamlined fabrication process developed by the authors in their previous works. I congratulate them on getting all aspect of the difficult experiment to work.

We thank Reviewer 2 for their kind words about our work.

In the end, however, I am not completely convinced, that the experiment at its current stage surpasses the high bar of novelty and demonstrates the technological leap forward justifying a publication in Nature.

We took this feedback to heart and feel that we may have written our prior draft too much as though this was simply two silicon qubits with a slightly different implementation. In fact, universal computing with only partial swaps is something which has never been done before in any platform, and is a highly distinct form of computing relative to both classical and existing spin-qubit implementations. We hope our revised draft which emphasizes this helps our work to pass the appropriately “high bar of novelty.”

Two-qubit gates have been implemented in many different platforms, including multiple quantum dot platforms and by many groups in the same SiGe platform, previously. There is therefore limited novelty in the concept itself.

It is important to emphasize that this is a universal *encoded* two-qubit gate. What has been done before is an exchange-based two-qubit gate for *individual* spins in similar but distinct SiGe platforms (e.g. overlapping gates rather than SLEDGE). The encoded two-qubit gate we present here is in a sense about 30 of these two-spin gates put together on 6 spins, but more importantly, it begins and ends in a fault-tolerant-friendly decoherence-free subsystem (DFS). GaAs multiqubit capacitive gates in two singlet-triplet qubits (four dots) are arguably comparable [12], but these require specially calibrated (or random) single-qubit gates using hyperfine gradients and are far more sensitive to charge noise. Other experimental analogies are the very recent implementation of fault-tolerant gates, also performed recently only in ions and superconductors [13, 14], perhaps the implementation of multiqubit logic in GKP states in superconducting resonators [15], and earlier encoded gates on more limited DFS systems in trapped ions [16]; certainly no demonstration like this has been previously published in a silicon system.

Furthermore, the method (gate-exchange) to perform the two-qubit gates does not use a new insight, but is a demonstration of what is known to be possible. In my opinion, the fabrication and technological feat of manufacturing and operating the array is what is the real meat of the paper. However, while still extremely impressive, it makes the manuscript perhaps better suited for a more technical journal.

It is true that “gate-exchange” is a ~ 25 -year old theoretical insight with prior proofs-of-principle, but universal exchange-only control was proposed well ahead of its time. It is rather normal for Nature to publish the culmination of decades of theoretical hope as an experimental reality.

I also considered the question of whether the paper demonstrated fundamentally new physics, functionality, or ideas from another discipline. Here, I think that two-qubit gates and SWAP operations based on exchange, singlet-triplet readout, in sub-ten quantum dot arrays have been reported before, and the authors have themselves demonstrated the SLEDGE architecture in previous manuscripts.

Rarely is a demonstration of this level of technology “fundamentally new physics,” but we emphasize that the universal computation with partial swaps is a novel form of computing not previously implemented in experiment, only in theory, and have revised our draft to emphasize this.

As for profound impact from a technological or scientific perspective, despite the impressive results, it is difficult for me to see how this architecture would form the building block of future scaled systems, especially as other groups have shown modular designs with a clear view towards going towards 2D. Perhaps

the manuscript could be strengthened with a clear statement about scaling of the architecture, or a citation to relevant proposals. It is especially difficult to see how the gate design here may extend to exchange coupling in the second dimension. To be most impactful at this stage of the spin qubit field, a qubit array on a two-dimensional chip should be able to demonstrate, or at least go towards, scalability and two-dimensionality.

We understand from this comment that we have not placed sufficient emphasis on the importance of the SLEDGE architecture used in our result. This technology was only quickly mentioned and cited, and, especially given Reviewer 2’s concerns about over-reliance on self-citation, its context was not well enough conveyed. To briefly answer the question here: the main innovation of SLEDGE is to separate the front- and back-end of the device using a multi-layer via process. This enables us to fabricate devices with multiple rails of quantum dots – “going towards 2D” – without modifying the rest of the process. The present device of course does not extend P dots into the second dimension, but only a straightforward extension of the fabrication process used to make it would be needed to enable it.

To remedy issue, we now include a new Fig. 2(d), which shows a vision of what the SLEDGE description indicates: the single gate metal layer that defines the qubits and the back-end wiring are separated in this architecture, allowing a straightforward stacking of vias to enable back-end wiring. This stacked-via approach is of course not original: it is precisely what enables 2D arrays of elements in present-day CMOS microprocessors. This lack of novelty is actually a feature, however, as compatibility with standard industry processes is the best path for true scalability.

We agree that such 2D arrays are critical for future scaling, and are aware of other proposals in the literature to achieve this goal. However, in our opinion, experimentally demonstrated platforms do not enable this type of extensible 2D BEOL wiring and cannot be considered to be modular. Planar multiqubit MOS devices [17] and all SiGe demonstrations published to date [1, 4, 9, 18], other than SLEDGE, use an overlapping gate architecture. This allows putting gates in a 2D “ring” (as we say above) but not to go to more complex topologies, simply because there is no way to route all the gates. There may be tricks to add extra gates to internal regions or to even attempt additional layers of overlapping gates, but these approaches are fundamentally not scalable. Similar problems occur for designs based on nanowires or fins [18, 19]. While cross-bar [20] and multiplexed [21] architectures in MOS have been proposed, these have not been demonstrated experimentally at the multiqubit level.

In summary, we feel that the scalability of our implementation is actually a major strength of our approach rather than a weakness. The fact that this was not clear to Reviewer 2 is extremely important feedback, which we have attempted to strongly rectify with changes to our text. We believe our architecture is the most extensible of any experimentally published platform in silicon, and we hope our added figure and discussion makes this clear. We do need to limit the scope of our paper to the novel demonstration which has been done, of course. Elaborations any further about how large 2D arrays need to be, how or whether to modularize

them, etc., are out of our scope to comment on here, but we have added some literature citations to some of the existing ideas for doing so in the new paragraph about 2D extensibility (paragraph 6).

I had some further comments on some specific questions: 1) For few-qubit arrays such as this, crosstalk is an important concern to address for the gate-defined quantum dot community. Is any calibration done to assess the effect of crosstalk (for example, single-qubit benchmarking applied only to the left triple-dot qubit, with a certain exchange pulse fidelity, followed by activating single qubit gates on the second qubit, showing no effect on exchange in the right two dots), or are the pulses and exchange gates tuned up concurrently, assuming the crosstalk remains more or less constant, and it is accounted for in this way? In my opinion, tuning up one set of pulses independently and then expecting the same effect when pulses on the other qubit are added in, is difficult. The authors only refer to this in the second page, stating that exchange gates benefit us by “natively providing low control crosstalk”, without any citation or evidence (while on page 4 they say that the pulses were not applied simultaneously, an important capability for a future quantum processor, precisely to avoid crosstalk).

This is a very important question worthy of significant elaboration. We previously left the topic of calibrating crosstalk mostly out of scope in our draft, largely for space and focus concerns, but of course we have previously put substantial thought into this question. Recognizing its importance, we now include an additional data set (Fig. Extended Data 2 (b), showing exactly the experiment Reviewer 2 describes) and lengthened discussion about the evaluation and measurement of crosstalk in our system (Supplement Section I.C). Still for reasons of space and focus, this discussion appears in the extended data and the supplement.

The key claim we make about crosstalk, discussed now in the supplement at greater length but worth reiterating here, is that it is extremely low *when pulses actuating different exchange energies are not “on” simultaneously*. It would certainly be possible to perform separate calibration routines for simultaneous pulsing, and could in principle improve gate fidelity by reducing total duration. However, we do not do that here, instead scheduling our control signals to assure that we never simultaneously engage two exchange interactions at once. As such, the demonstration of low crosstalk is to show that when we interleave encoded single-qubit gates on one encoded qubit with randomized benchmarking on the other encoded qubit, we see no additional error relative to not doing so (e.g. a nearby gate has the same effect as just idling). For scaling, one eventually needs to worry about truly *simultaneous* pulses; for devices not much larger than the present one, we anticipate (but have not shown) that the physical distance and field screening between gates can render the associated capacitive crosstalk negligible.

To answer the direct question: we calibrate each exchange axis independently of one another and schedule them together without any additional calibration. We appreciate that Reviewer 2 recognizes that this is difficult! The “contextual” calibration Reviewer 2 describes, where the value of one pulse depends on its immediate history, is not necessary here because of our use of an idle period

between operations. Since exchange is exponentially suppressed on idling qubits, crosstalk of control pulses to neighbors will not meaningfully affect the experiment (for example, taking the off-state J from 1 KHz to 2 KHz for 10 ns does not cause a measurable error.) The idle duration also gives our voltage waveforms sufficient time to decay down to a negligible level before a new pulse is applied.

2) Figure 1 is compelling and well designed. However, the materials making up the various components in the top-view and cross section are not described well either in a legend or in the caption. This should be rectified as I had to dig into the text and other papers to understand the design, especially the patch in the centre of 1(a).

We now indicate these materials in the caption (Fig. 2(a-b)), although details still require the space of a fully distinct citation on fabrication. We also indicate in the Fig. 2 (a) caption that “Unlabeled patches of metal are field gates, held at constant bias.”, to help with this single question without having to look up the reference.

3) Overhead: The number of exchange pulses required to perform a two-qubit gate are many, with probably some probability of error per pulsed operation; this way of encoding qubits therefore seems to come with immense overhead and calibration/operational complexity. The authors suggest in the supplementary (page 15) to use magnetic decoupling pulses and in page 5 of the main text to use gate set tomography to improve their fidelities (by the way, they should cite Kuemmeth, UCPH (Nat. Nano , 2017) and Bluhm, Aachen (Nat. Comm 2017) for decoupling experiments and gate set calibration experiments). These further sequences will add even more complexity to the protocol for even a single two-qubit gate. The number of pulses required means that (Figure S3) a single Fong-Wandzura CNOT (without decoupling pulses) takes about 400ns to perform, while the idle singlet state $T2^*$ is stated as 3.5us. The LC-CNOT is 46 pulses, and the LCCZ is 44 as well. This seems like a pretty bad scaling factor for performing multiple gates before decoherence; can the authors comment on this overhead?

This is a very important question – thank you for giving us an opportunity to respond to it.

First, we would argue that the “overhead” here is not so immense when compared to other technologies. Comparing first in gate duration and taking as reference the recent Xue paper [2], single-qubit rotations performed with EDSR for single spins typically take 100-200 ns and CZ primitives 100 ns. It is true that our two-qubit entangling operation is somewhat slower than that, but our single-qubit gates are much faster; a quantum circuit would be composed of many of both. (The comparison to superconducting qubit gates is not so straightforward; prominent examples in the literature range from tens of ns to a few microseconds. There are also important trades between excited state occupation, RF crosstalk, off-state coupling rates, and so on.) So, in the dimension of time, which as Reviewer 2 implies, matters most for decoherence, there is not really substantial overhead relative to this other technology.

From the perspective of control complexity, we also think the challenge is not so significant. We calibrate a continuous transfer function of J to voltage for each axis, so really we only need to calibrate five things and then query those interpolatable functions for whatever angle our gates call for. (As we say in the previous response, each pulse is *not* calibrated in context; the voltage amplitude for a π pulse on some particular axis is the same regardless of what plays before or after it.) While there has been a significant amount of work in developing this process, the calibration methods used in other technologies are also highly nontrivial and rightfully sources of pride for their operators.

The amount of data needed to program our entangling control pulses are also comparable to other technologies. Taking the example of the Xue et al. [2], their 100-200 ns single-qubit RF pulses likely use a combination of single sideband modulation and amplitude modulation. This means they need to program two AWG channels (for the I and Q quadratures) every sample period (say, 1 ns) with a unique DAC value. This comprises approximately 400 DAC words, which somewhat *more* than we would need for our CZ gate.

Second, the most important question is not how many pulses a control sequence takes or the wall-clock time it takes to execute, but instead its fidelity. Our two-qubit Clifford gates are the same or better as recent quantum dot results and also compare favorably to many superconducting and ion results. This is despite the fact that we use more than $10\times$ more pulses than the single-spin exchange-entangled qubits; the lack of magnetic field gradients and the sophistication of our devices and calibration routine pay big dividends here. There is also a clear path to improving gate fidelity in our technology: for example, increasing speed or reducing nuclear isotope concentration, as we discuss in the paper. These are expensive and time-consuming directions, but have no fundamental obstacles relative to physics or even novel engineering.

Third, we think that even if our system were meaningfully more complex than other technologies to calibrate (which we hope you agree is not so clear, given the discussion above), its advantages far outweigh those costs. The complexity in calibration and managing a linear pre-factor of dots and pulses is largely a concern with software development. While the solution to this is certainly an interesting topic and will take skillful engineering, it is not expected to be a major scientific challenge. However, comparing the complexity of instructions for exchange-only control to the complexity of maintaining RF local oscillators in a scaled machine varies is a *very* interesting question, which we believe strongly favors the exchange-only, baseband approach, but a full argument thereof is beyond the scope of the present demonstration.

Concerning decoupling: decoupling for the exchange-only system is a rich topic. Sequences for doing so in the exchange-only system were proposed by our group a decade ago [22] and their demonstration is the topic of a forthcoming manuscript which we cite [23]. Performing two-qubit gates while simultaneously decoupling has also been proposed [24]. Note that these are necessarily more sophisticated pulse sequences than the Kuemmeth and Bluhm papers indicated by the reviewer, which are (1) in GaAs, (2) on double-dots without full-qubit control except via the hyperfine interaction which is decoupled in the demonstrations. Decoupling for

an encoded exchange-only qubit is more complex, and truly a topic for a different paper; discussion or citation list of the many, many papers about decoupling would take far too much space for such an out-of-scope (but still important) topic. Our response to this comment is, in fact, to remove the statement that decoupling would improve our fidelity (as it does take substantially more pulses, such a claim is not substantiated either by our data or by the theory mentioned above). Our supplement Section II now only indicates that the SWAP gate has some decoupling characteristics, for which the theory of Ref. 22 and our simulations support, as well as our data to the extent that we can measure the SWAP fidelity. (IRB of a gate with substantially better fidelity than the two-qubit Clifford is inaccurate, resulting in the high error bar on the decoupling-SWAP fidelity.)

Also, we did not write that “gate set tomography” would “improve our fidelity;” arguably it could increase our categorization of the errors contributing to our fidelity, but adapting GST to our encoded system including gauge and leakage is ongoing work.

5) **Noise analysis** The noise analysis, in my opinion, came across as the weakest point of an otherwise strong manuscript. I was unable to convince myself that magnetic noise is indeed the limiting factor. Figure 4e seemed to me to show that magnetic field does not in fact affect Clifford fidelities much until at fields $>3\text{mT}$ the superconducting gate stack becomes involved. The papers cited in support of the analysis that magnetic field is the main contribution to the error (Ref 1, 22, 34) seem to all be self-cites. More varied citations, perhaps to theory by other groups, or a more clear model or demonstration of the errors from charge noise vs magnetic noise is required, I think, to make the statement the authors make that in their system “The dominant error source of all entangling gates is hyperfine magnetic noise”. Lastly, if the magnetic noise hypothesis were to be true, the authors suggest that the SiGe layers on either side of the quantum well may need to be isotopically purified, which perhaps also impacts scaling? How far from the quantum well does the wafer need to be isotopically purified?

We have perhaps had a miscommunication here which we now attempt to rectify in our revised draft. By “magnetic noise,” for our exchange-only system, we are referring to absolutely any source of locally varying magnetic field, as witnessed by the electron spins. By “locally”, we mean fields that change on the scale of the inter-dot spacing, since uniform magnetic fields by construction do not affect a DFS-encoded qubit, even if they fluctuate in time. For fields less than 3 mT, where most results are shown, there is only one such source of “magnetic noise,” and that is the hyperfine fields of randomly oriented ^{29}Si and ^{73}Ge nuclei. We have made very detailed models of this noise, as well as charge noise, but we did not elaborate these models in the present manuscript, since they are the same as in our previous works [6, 25]. Due to considerations of space and focus, we asked the reader to rely on these descriptions for an elaboration. We understand that this did not succeed in conveying our results in our previous draft, and so we now more fully elaborate on those error models and simulations in Supplement Section I, largely recapitulating previously published work, again in the supplement (as

we do not want to detract from the novel work in the space-limited main text.)

The reason “self-cites” are relied upon here is, plainly, due to the novelty of our approach. Although there is significant literature on T_2^* and hyperfine noise in Si/SiGe systems, single-spin approaches built in large magnetic field gradients significantly change both the magnetic and charge noise landscape. (Even more literature exists for GaAs, but here the 100% abundance of spin-3/2 nuclei constitutes a set of different issues.) As such, prior literature results have only limited relevance and, we believe, would cause more confusion than help in elaborating our noise. Nonetheless, with the freedom of space in the supplement, we have also added a discussion of this prior literature in Section I.A of the supplement, again largely summarizing the discussion present in some of our other papers.

To the reviewer’s question, “how far from the quantum well does the wafer need to be isotopically purified”, this distance is set by the tail of the vertical wavefunction of the electron in the quantum dot, which is quite small: a few nanometers should suffice.

6) Leakage: It is known that the EO qubit gains all-electrical control but makes the sacrifice that it is more vulnerable due to a larger leakage subspace. The authors state that they are not completely sure about the attribution of leakage error versus encoded errors via randomised benchmarking. I think this goes along with point (5) in that it is difficult to parse (after reading the entire paper a few times) what is actually the top limiting factor for the two-qubit fidelity, and whether the authors know this clearly, and how they hope to tackle it next. Is the limitation indeed the magnetic noise and is leakage not a problem? In the outlook they seem to intend to work on it in the future to reach fault tolerance.

The limiting factors for two-qubit fidelity is overwhelmingly hyperfine noise, with the next-largest term being charge noise, which we model to contribute 6% of the budget. We believe we have stated this clearly, but the reviewer’s confusion seems to be about the words “magnetic noise” and “leakage.” “Magnetic noise” and “leakage” are not in competition with one another: magnetic noise is a physical error source which causes both encoded and leakage error. (“Charge noise” is another physical error source which causes both encoded and leakage error, at least for the case of interactions between two or more encoded DFS qubits.) This is perhaps a confusion when one tries to cast the encoded qubit into the language of unencoded qubits (which can also have something called “leakage,” which is quite different.) We have attempted to clarify our language in this regard throughout the present revision, by more precisely defining “leakage” in the introduction and putting more care into the use of the term “magnetic noise” in both the main text and supplement.

We note that our error models unambiguously predict what the leakage error is *expected* to be, and our data are consistent with those models. We are quite confident about the amount of leakage vs. encoded error due to magnetic noise via simulation. (The relative rates of leakage to encoded error turn out to be very similar to the three-dot case [25].) However, we remain careful to avoid claiming that our methods of measurement are able to experimentally confirm this separation. *Experimentally* separating the total gate error into “encoded

error,” that is, error which occurs within the encoded subsystem, and “leakage error,” that is, error which leaves spins in unencoded states, remains a challenge with two encoded qubits, but the presence of both and their relative contributions are not a source of uncertainty, and both are assuredly summed together in the randomized-benchmarking-derived fidelity we use. In an earlier demonstration with a single DFS qubit (e.g. three spins) [25], we proposed an elegant solution for sorting the error types for the case of single qubit rotations, which is the premise of the “blind RB” protocol. Unfortunately, it is not trivial to extend this protocol into two encoded qubits, nor can we do process tomography over the extended system of both encoded states and leakage states. So the challenge is not to characterize total fidelity but rather the sorting of that error into encoded vs. leakage errors. This is a subtle, quantum-information-related intricacy that arises with the novelty of encoded operation; it has not encountered for single-spin or transmon qubit types.

6) I also had some comments about referencing: the paper has quite a few self-cites: 9, including authors at the same institution. They should also, or instead, cite other groups on demonstrating spin qubit arrays (especially 2D arrays, such as Germanium in Delft and GaAs in Delft, Copenhagen, Grenoble), gate set tomography (Aachen), and symmetric operation; this would also bolster confidence in the section on noise analysis (especially in the last two paragraphs before the conclusion).

As we mentioned above, our qubit is novel compared to other quantum dot academic or industrial efforts, which necessarily means we will need to lean on our own earlier work to explain the details of the present result. (It may be worth emphasizing that this result is the culmination of the years of research reported in those self-citations.) There are a few other examples of experimental DFS encodings in the literature, but they are not in silicon and are only superficially relevant. Given limited space, we chose our citation list based on those papers most informative to our approach.

We are not attempting to comprehensively acknowledge all approaches to 2D scaling or material choice in our manuscript, especially given its new focus on SWAP-based logic rather than comparisons to other spin qubits. Delving into comparative methods to make 2D arrays or other materials is also a slippery slope on citations – why not 2D arrays of transmons or neutral atoms? Similarly, why are single holes in Ge relevant, given their different physics and control schemes, other than their elemental composition? Nonetheless, we have attempted to include the citations indicated into the revised text. We hope the editors allow the expanded citation count, and if they do not, please note that one of us (TDL) is a co-author of a recent review of spin qubits [26] which has hundreds of such citations, which of course we cite, enabling the interested reader to find a large variety of papers on Ge, GaAs, GST, etc.

REFeree 3

The encoded qubit scheme for quantum dot spin qubits has been proposed and explored for over two decades ...Since exchange interactions can be easily controlled electrically by tuning the gate voltages on the metallic leads, it provides a viable route for scalable quantum computers. The authors previously demonstrated single-qubit gate operations and now add two-qubit gate operations to complete the universal set of gates. It is an important step for encoded qubit quantum computers, and also shows the capability of current state-of-the-art multiple QD devices. The reported gate fidelities are not yet at the surface code threshold level, but at a high 90's % with some room for improvements.

We're delighted that the reviewer sees the value of the approach, and want to re-emphasize that "tuning gate voltages on metallic leads" is a key advantage because it is *only* that, and, more over, those voltage pulses can be asynchronous and unshaped. When exchange-only was proposed 20 years ago, its noise-cancelling advantages were of course advertised, but its engineering advantages were likely not fully appreciated. We hope this manuscript appeals those who knew the fault-tolerance advantages from the early theory in that it also helps elaborate technical implementation advantages.

On the other hand, although a number is not given in the manuscript, the SPAM errors still seem to be significant.

Especially on one side of this device, SPAM is not as strong on this device as others, largely because of the size of quantum well chosen (e.g. 5 nm). We have cited a new paper which quantitatively evaluates SPAM, as well as a study of SPAM-limiting valley splitting numbers of well sizes [5, 8]. That paper continues the conversation of how SPAM can be improved in future devices (for example, narrower wells) to further develop this technology. The main technical result we present here is in exchange-based gate control, for which only one "good" measurement channel was needed. The unfortunate fidelity of half of this particular device was inconvenient, but in no way affects the soundness of our claims. Future results that depend on multiple measurement channels will certainly need to incorporate even more advances in device design that were not strictly necessary here. If the technology is being evaluated as a whole, we would recommend the Blumoff, et al. [5] which focuses on SPAM and introduces some innovations that could be incorporated into later demonstrations.

Another important advancement is leakage-controlled gate operations. Encoding a qubit state in a larger quantum system introduces the possibility that the quantum system can be outside of the encoded qubit subspace, making it vulnerable to leakage errors. Although the leakage-controlled gates did not lead to higher fidelity (probably because it requires a longer sequence of exchange control pulses), it may be essential for further development of encoded qubit devices.

Yes, unfortunately this important point is necessarily de-emphasized in our revised presentation as we have focused more on the initial novelty of the whole premise of computation by partial swaps. The leakage-controlled gate operation is, however, a completely new construction to this manuscript and very important by itself. As we have indicated, its leakage control aspects are understood theoretically but are not strongly indicated by our experiment (as our metric has no direct sorting of leakage error), which is a further reason to deemphasize this nonetheless “important advancement.” We nevertheless greatly appreciate Reviewer 3’s recognition.

I think this manuscript is well written and it warrants publication in Nature.

We thank you for your support of our effort!

1. SPAM error seems to be quite large yet. What are the main causes and what will be a plausible strategy to improve it?

Do see our response to Reviewer 1 on SPAM compared to other silicon qubit systems; in fact for the M1 side of our device, we see 99% RB contrast, indicating a much lower SPAM error than all silicon qubit publications to date. The M2 side is on par with most publications but inferior to M1. The main cause is inadvertent population in valley states, which have largely to do with the construction of the quantum well. We have other six-dot devices published with $> 99.7\%$ SPAM fidelity, and HRL’s epitaxy has been used for $> 99\%$ -fidelity SPAM devices in other groups [4], so the strategy for improvement is to, in the future, combine this epitaxy with the SLEDGE gate architecture, as we now more clearly indicate in the text (paragraph 8: “Valley splitting may be more frequently high in narrower wells [8], or possibly with other mitigations [9], and indeed SPAM fidelity was validated to reach over 99.7% in a similar, narrower-well device in Ref. 5.”).

2. It is briefly mentioned that the exchange pulses were applied sequentially to limit crosstalk. I thought one of the main advantages of the exchange-only encoded qubit is that all controls are done by simple voltage controls and the crosstalk is not a significant issue once calibrations are done. If crosstalks limit the parallel application of voltage pulses, then it may be a serious problem for the future development of larger systems. Any comments on this?

Yes, please see our response to reviewer 2 on this question, our extended discussion in the supplement Section I.C, and the new dataset provided (Fig. Extended Data 2b). To briefly reiterate, simultaneous operation without extra calibration should be possible, but not quite in devices as small as this one.

3. The exchange control is vulnerable to charge noise and it is beneficial to perform the operations on a sweet spot or symmetric point. In this multiple quantum dot device, how the sweet spots are determined experimentally? Can parallel applications of exchange pulses be done while maintaining the sweet spots, as is expected in theory (Shim and Tahan, PRB 93, 121410(R), 2016)?

It is important to recognize that our approach uses pairwise exchange only, and each pairwise exchange uses a sweet-spot. Exchange is “off” for all other spins whilst one pair at a time undergoes a partial swap. In this sense, we are always using sweet spots. The experimental determination of those sweet spots is done, for each axis, exactly as described in [27], as we cite in the present manuscript rather than copy.

The theory of Shim and Tahan and related papers have to do with applying exchange to three dots at a time. This is much more difficult to calibrate and as such, although the physics of our device allows it, our software and operation procedures would require a radically different calibration procedure which no one has yet demonstrated or published. As an example, we can currently make strong assumptions about the Bloch sphere axis about which a given exchange pulse is rotating; when multiple exchange energies are large at a time, that assumption is broken. Theoretical simulations, which we do not report here, suggest no significant gain in charge noise performance from this type of sweet-spot relative to our present operation, once the full physics of dots (including wave-function distortions captured by FCI and disorder are accounted for.) However, the existence of such sweet spots is especially important if seeking to couple a triple-dot to a microwave resonator, or doing capacitive gates (or both), either of which may be important for connecting limited-scale arrays. See also Pan et al. [28] in which we elaborate this interesting line of discussion.

4. For two-qubit operations between two exchange only qubits, there is a scheme that requires a much shorter sequence of pulses based on the difference in strengths in intra-qubit and inter-qubit exchange interactions (Doherty and Wardrop, PRL 111, 050503, 2013). Is it plausible in this device?

This has effectively the same answer as the previous question. It is plausible, but very challenging to calibrate, and that calibration is not within the operating procedures described in our manuscript. Moreover, such an implementation would be more subject to charge noise, and not have a clear speed advantage (or other advantage relative to magnetic noise.) A key resource we employ here is the Fong-Wandzura construction, which allows us to not polarize two of the six spins in our array. The Doherty and Wardrop proposal, like the DiVincenzo proposal [29], would require somehow polarizing these spins, which would require an element no longer in the “rules” (e.g. available resources) of exchange-only, such as a deliberate field gradient, a spin-polarized tunnel barrier, or the use of single-spin relaxation. The Doherty and Wardrop proposal can be modified to be, like Fong-Wandzura, independent of gauge polarization [see Ladd and Jones, Ref. 30]; this publication also indicates some of the noise trade-offs of the two strategies.

5. Other than the larger number of physical qubits required, what do you see as the biggest hurdle for scaling up these encoded qubit devices?

We identify two primary challenges to scaling DFS EO qubits: a technical one, fabrication and disorder; and a social one, engagement and acceptance by the

larger quantum information community.

The difficulty and cost of a high-yielding quantum dot process is the most straightforward story. Although we indicate in Fig. 2d that improved BEOL wiring can enable 2D arrays, the actual development of such a process is akin to the processes developed for CMOS microprocessors. From a scientific point of view, this is a positive: such development is possible and will succeed, given requisite resources. The amount of time it would take to develop such a process at high yield is also basically known, and is considerable. Every additional layer incrementally slows the learning process; although the end result is nearly the same as the BEOL of commodity microprocessor chips today, we cannot simply peel off the BEOL from an existing chip and glue it to a qubit chip: the process development must be redone. The devices must also have sufficiently low disorder to be relied upon for larger quantum circuits. Not only do electrons need to be reliably loaded in a sufficient number of quantum dots, but those spins need to be controllable with sufficiently high fidelity that they constitute useful resources for computing. As we mention in the manuscript, part of attaining this “parametric yield” is at least scientifically straightforward, e.g. acquiring more highly enriched materials for the quantum well to mitigate magnetic noise. However, we anticipate that even assuming those improvements, significant gains will need to be made in other areas like valley excited state energy and charge noise. We see no fundamental barrier in either case, however — it is “just” a matter of process development and design innovation.

The second challenge is, frankly, convincing the larger quantum information community that EO DFS qubits are worth studying. In a world where the accepted standard for technological progress is the number of qubits and fidelity with which they can be controlled, our approach faces significant headwinds. As an example, the device we report on could otherwise be treated as a six-qubit device if operated in a more conventional way (e.g. as Loss-DiVincenzo qubits). The fidelity of the entangling operations, assuming that they were not significantly degraded by whatever modification was made to incorporate RF addressability, would also be considerably higher than the current state of the art. (Our per-exchange-pulse infidelity, inferred from either single- and two-qubit operations, is between 0.04% and 0.07%. This is more than 10 times better than the recent *Nature*-published results that use a single exchange pulse for a CZ gate.) We are instead operating it as *two* encoded qubits that amplify any exchange error by 40 times, hurting the effective two-qubit metric. As we have hopefully communicated, we strongly believe this trade is worthwhile in the context of the ultimate goal of fault tolerance, but that argument is subtle. Getting more researchers to develop DFS EO qubits is critical to the technology’s ultimate success, in order to solve scientific problems like the ones mentioned above. It is thus incumbent on us to convince the community that this trade is at least worth entertaining, and getting their engagement. The exposure we’d enjoy from publishing this result in *Nature* would be extremely helpful on this front, which is a big part of why we are taking the time to write this appeal rather than accepting publication in a more

specialized journal.

- [1] A. Noiri, K. Takeda, T. Nakajima, T. Kobayashi, A. Sammak, G. Scappucci, and S. Tarucha, Fast universal quantum gate above the fault-tolerance threshold in silicon, *Nature* **601**, 338 (2022).
- [2] X. Xue, M. Russ, N. Samkharadze, B. Undseth, A. Sammak, G. Scappucci, and L. M. Vandersypen, Quantum logic with spin qubits crossing the surface code threshold, *Nature* **601**, 343 (2022).
- [3] M. T. Madzik, S. Asaad, A. Youssry, B. Joecker, K. M. Rudinger, E. Nielsen, K. C. Young, T. J. Proctor, A. D. Baczewski, A. Laucht, V. Schmitt, F. E. Hudson, K. M. Itoh, A. M. Jakob, B. C. Johnson, D. N. Jamieson, A. S. Dzurak, C. Ferrie, R. Blume-Kohout, and A. Morello, Precision tomography of a three-qubit donor quantum processor in silicon, *Nature* **601**, 348 (2022).
- [4] A. R. Mills, C. R. Guinn, M. J. Gullans, A. J. Sigillito, M. M. Feldman, E. Nielsen, and J. R. Petta, Two-qubit silicon quantum processor with operation fidelity exceeding 99%, arXiv:2111.11937 (2021).
- [5] J. Z. Blumoff, A. S. Pan, T. E. Keating, R. W. Andrews, D. W. Barnes, T. L. Brecht, E. T. Croke, L. E. Euliss, J. A. Fast, C. A. C. Jackson, A. M. Jones, J. Kerckhoff, R. K. Lanza, K. Raach, B. J. Thomas, R. Velunta, A. J. Weinstein, T. D. Ladd, K. Eng, M. G. Borselli, A. T. Hunter, and M. T. Rakher, Fast and high-fidelity state preparation and measurement in triple-quantum-dot spin qubits, arXiv:2112.09801 (2022).
- [6] J. Kerckhoff, B. Sun, B. Fong, C. Jones, A. Kiselev, D. Barnes, R. Noah, E. Acuna, M. Akmal, S. Ha, J. Wright, B. Thomas, C. Jackson, L. Edge, K. Eng, R. Ross, and T. Ladd, Magnetic gradient fluctuations from quadrupolar ^{73}Ge in Si/SiGe exchange-only qubits, *PRX Quantum* **2**, 010347 (2021).
- [7] O. E. Dial, M. D. Shulman, S. P. Harvey, H. Bluhm, V. Umansky, and A. Yacoby, Charge noise spectroscopy using coherent exchange oscillations in a singlet-triplet qubit, *Phys. Rev. Lett.* **110**, 146804 (2013).
- [8] E. H. Chen, K. Raach, A. Pan, A. A. Kiselev, E. Acuna, J. Z. Blumoff, T. Brecht, M. D. Choi, W. Ha, D. R. Hulbert, M. P. Jura, T. E. Keating, R. Noah, B. Sun, B. J. Thomas, M. G. Borselli, C. Jackson, M. T. Rakher, and R. S. Ross, Detuning axis pulsed spectroscopy of valley-orbital states in Si/Si-Ge quantum dots, *Phys. Rev. Appl.* **15**, 044033 (2021).
- [9] T. McJunkin, B. Harpt, Y. Feng, M. Losert, R. Rahman, J. P. Dodson, M. A. Wolfe, D. E. Savage, M. G. Lagally, S. N. Coppersmith, M. Friesen, R. Joynt, and M. A. Eriksson, SiGe quantum wells with oscillating ge concentrations for quantum dot qubits, arXiv:2112.09765 (2021).
- [10] T. I. Baturina, A. Y. Mironov, V. M. Vinokur, M. R. Baklanov, and C. Strunk, Localized superconductivity in the quantum-critical region of the disorder-driven superconductor-insulator transition in tin thin films, *Phys. Rev. Lett.* **99**, 257003 (2007).
- [11] W. Ha, S. D. Ha, M. D. Choi, Y. Tang, A. E. Schmitz, M. P. Levendorf, K. Lee, J. M. Chappell, T. S. Adams, D. R. Hulbert, E. Acuna, R. S. Noah, J. W. Matten, M. P. Jura, J. A. Wright, M. T. Rakher, and M. G. Borselli, A flexible design platform for si/sige exchange-only qubits with low disorder, *Nano Lett.* **22**, 1443 (2021).
- [12] M. D. Shulman, O. E. Dial, S. P. Harvey, H. Bluhm, V. Umansky, and A. Yacoby,

- Demonstration of Entanglement of Electrostatically Coupled Singlet-Triplet Qubits, *Science* **336**, 202 (2012), publisher: American Association for the Advancement of Science.
- [13] S. Rosenblum, P. Reinhold, M. Mirrahimi, L. Jiang, L. Frunzio, and R. J. Schoelkopf, Fault-tolerant detection of a quantum error, *Science* **361**, 266 (2018).
 - [14] L. Postler, S. Heußen, I. Pogorelov, M. Rispler, T. Feldker, M. Meth, C. D. Marciniak, R. Stricker, M. Ringbauer, R. Blatt, *et al.*, Demonstration of fault-tolerant universal quantum gate operations, *Nature* **605**, 675 (2022).
 - [15] P. Campagne-Ibarcq, A. Eickbusch, S. Touzard, E. Zalys-Geller, N. E. Frattini, V. V. Sivak, P. Reinhold, S. Puri, S. Shankar, R. J. Schoelkopf, *et al.*, Quantum error correction of a qubit encoded in grid states of an oscillator, *Nature* **584**, 368 (2020).
 - [16] T. Monz, K. Kim, A. S. Villar, P. Schindler, M. Chwalla, M. Riebe, C. F. Roos, H. Häffner, W. Hänsel, M. Hennrich, and R. Blatt, Realization of universal ion-trap quantum computation with decoherence-free qubits, *Phys. Rev. Lett.* **103**, 200503 (2009).
 - [17] R. C. Leon, C. H. Yang, J. C. Hwang, J. Camirand Lemyre, T. Tanttu, W. Huang, J. Y. Huang, F. E. Hudson, K. M. Itoh, A. Laucht, *et al.*, Bell-state tomography in a silicon many-electron artificial molecule, *Nature communications* **12**, 1 (2021).
 - [18] A. M. J. Zwerver, T. Krähenmann, T. F. Watson, L. Lampert, H. C. George, R. Pillarisetty, S. A. Bojarski, P. Amin, S. V. Amitonov, J. M. Boter, R. Caudillo, D. Corras-Serrano, J. P. Dehollain, G. Droulers, E. M. Henry, R. Kotlyar, M. Lodari, F. Luthi, D. J. Michalak, B. K. Mueller, S. Neyens, J. Roberts, N. Samkharadze, G. Zheng, O. K. Zietz, G. Scappucci, M. Veldhorst, L. M. K. Vandersypen, and J. S. Clarke, Qubits made by advanced semiconductor manufacturing, arXiv:2101.12650 (2021).
 - [19] T. Bédécarrats, B. C. Paz, B. M. Diaz, H. Niebojewski, B. Bertrand, N. Rambal, C. Comboroure, A. Sarrazin, F. Boulard, E. Guyez, J. M. Hartmann, Y. Morand, A. Magalhaes-Lucas, E. Nowak, E. Catapano, M. Cassé, M. Urdampilleta, Y. M. Niquet, F. Gaillard, S. De Franceschi, T. Meunier, and M. Vinet, A new FDSOI spin qubit platform with 40nm effective control pitch, in *2021 IEEE Int. Electron Devices Meeting* (2021) pp. 1–4.
 - [20] R. Li, L. Petit, D. P. Franke, J. P. Dehollain, J. Helsen, M. Steudtner, N. K. Thomas, Z. R. Yoscovits, K. J. Singh, S. Wehner, *et al.*, A crossbar network for silicon quantum dot qubits, *Science advances* **4**, eaar3960 (2018).
 - [21] L. M. K. Vandersypen, H. Bluhm, J. S. Clarke, A. S. Dzurak, R. Ishihara, A. Morello, D. J. Reilly, L. R. Schreiber, and M. Veldhorst, Interfacing spin qubits in quantum dots and donors—hot, dense, and coherent, *npj Quantum Information* **3**, 1 (2017), number: 1 Publisher: Nature Publishing Group.
 - [22] J. R. West and B. H. Fong, Exchange-only dynamical decoupling in the three-qubit decoherence free subsystem, *New J. Phys.* **14**, 083002 (2012).
 - [23] B. Sun *et al.*, Full-permutation dynamical decoupling in triple-quantum-dot spin qubits, In Preparation (2022).
 - [24] F. Setiawan, H.-Y. Hui, J. P. Kestner, X. Wang, and S. D. Sarma, Robust two-qubit gates for exchange-coupled qubits, *Phys. Rev. B* **89**, 085314 (2014).
 - [25] R. Andrews, C. Jones, M. Reed, A. Jones, S. Ha, M. Jura, J. Kerckhoff, M. Leventorff, S. Meenehan, S. Merkel, A. Smith, B. Sun, A. Weinstein, M. Rakher, T. Ladd, and M. Borselli, Quantifying error and leakage in an encoded Si/SiGe triple-dot qubit, *Nat. Nanotechnol.* **14**, 747 (2019).
 - [26] G. Burkard, T. D. Ladd, J. M. Nichol, A. Pan, and J. R. Petta, Semiconductor spin

- qubits, arXiv:2112.08863 (2021).
- [27] M. D. Reed, B. M. Maune, R. W. Andrews, M. G. Borselli, K. Eng, M. P. Jura, A. A. Kiselev, T. D. Ladd, S. T. Merkel, I. Milosavljevic, E. J. Pritchett, M. T. Rakher, R. S. Ross, A. E. Schmitz, A. Smith, J. A. Wright, M. F. Gyure, and A. T. Hunter, Reduced sensitivity to charge noise in semiconductor spin qubits via symmetric operation, *Phys. Rev. Lett.* **116**, 110402 (2016).
 - [28] A. Pan, T. E. Keating, M. F. Gyure, E. J. Pritchett, S. Quinn, R. S. Ross, T. D. Ladd, and J. Kerckhoff, Resonant exchange operation in triple-quantum-dot qubits for spin-photon transduction, *Quantum Sci. Technol.* **5**, 034005 (2020).
 - [29] D. P. DiVincenzo, D. Bacon, J. Kempe, G. Burkard, and K. B. Whaley, Universal quantum computation with the exchange interaction, *Nature* **408**, 339 (2000).
 - [30] T. D. Ladd and C. Jones, Gauge-invariant adiabatic two-qubit gates for exchange-only qubits, arXiv:1712.05849 (2017).

Reviewer Reports on the First Revision:

Referee #1 (Remarks to the Author):

Dear Editor,

Thank you for forwarding me the rebuttal letter from the authors of the paper, Universal Logic with encoded spin qubits in silicon. I appreciate the effort the authors have put in trying to answer my original concerns about the suitability of this manuscript for Nature. After a careful read of the responses and modified manuscript, my impression stands. Let me elaborate.

The authors have responded to my major concern related to published work in Nature on two qubits gates in silicon. The authors discussed that two-qubit Clifford gates should be utilized as the metric for comparison. This could be the case but even if their preferred metric was utilized the results are on a par with published work. When focusing on SPAM errors, although the manuscript now presents results showing 99% contrast in single-qubit randomized benchmarking, and I appreciate the addition (although similar results are published in [5]), this is an incremental improvement over ref [4] and on a par with published work [Nature 609, 919 (2022)]. Further, there are now two published works [Nature 609, 919 (2022)] and [Nature 608, 682 (2022)], that show an increased number of qubits and three-qubit algorithms.

Overall, it is difficult for me to conclude that the novelty of this modified manuscript should warrant publication in Nature. As in my original response, this should be by no means taken as a criticism to the outstanding technical prowess demonstrated in the text. Ultimately this paper should be published in a high impact venue.

Allow me to make a final comment related to new responses: Could the authors comment on the new Fig.2 d on the feasibility of maintaining the same interconnect footprint as one moves up in the metal layers of the SLEDGE architecture? Could the authors extend on how readout will happen, i.e. where the sensor will be located?

Overall, the rest of the responses are satisfactory.

Best regards,

Referee #2 (Remarks to the Author):

I have now reread the response, manuscript changes, and while the authors have answered a few of the technical questions, I am afraid my opinion is fundamentally unchanged.

1) I very much appreciated the added discussion of crosstalk, as I feel it strengthens the paper. However, the authors describe crosstalk as not a problem due to the high on-off nature of the exchange interaction, but then say that they were just unable to perform simultaneous operation in

this flagship experiment, and had to compensate by staggering pulses slightly. Point 3 of Reviewer 1 also refers to this, I believe, and the answer to me is not satisfactory. They say on page 4 of the response that they perform "interleaving RB on one encoded qubit with quantum action on a second, showing no adverse effect of the parallel operation" as an argument for their low-crosstalk approach again; it is yet again unclear (even in the caption of the figure!) if "parallel" means "simultaneous" or if the pulses are again slightly staggered, because they then say in their response to point 4 of reviewer 1: "The distance at which simultaneous control pulses could safely be applied without meaningful error is somewhat larger than the present device, so we do not employ simultaneous control here." and to me "However, we do not do that here, instead scheduling our control signals to assure that we never simultaneously engage two exchange interactions at once." The reviewer clearly meant "parallel" in the sense of "simultaneous" here. I think this is dangerous, and the authors should be very clear at all times, since most people in the field would understand crosstalk during qubit control pulses to refer to simultaneous operation.

2) The authors in their response to me now mention that they would reformulate the impact of their manuscript as: "universal computing with only partial swaps is something which has never been done before in any platform". It is unclear whether this "first" is really something much of the scientific community has been working towards a demonstration of. I am therefore unsure if it strengthens the paper.

3) Similarly, that this is an encoded two-qubit gate does not inherently make it better. The authors say: "It is important to emphasize that this is a universal encoded two-qubit gate. What has been done before is an exchange-based two-qubit gate for individual spins in similar but distinct SiGe platforms"

The right way to go about claiming an advantage for encoded qubits would be to demonstrate a coherence advantage, or to go beyond the state of the art for non-encoded qubits, and not to demonstrate that gating is now more complex but still achievable. They also alter their previous statement of "never been done before in any platform" to be "certainly no demonstration like this has been previously published in a silicon system" after providing some examples of fault-tolerant gating in other systems.

4) My comment about the second dimension was not based on any misunderstanding of the fabrication, or that the authors have "not placed sufficient emphasis on the importance of the SLEDGE architecture" as they say; I am on the contrary familiar with it. I repeat that without a demonstration it is still not clear what method the authors plan to use to substantiate their claim that they could have demonstrated, for example, the current experiment in both the x and y dimensions. Simply BEOL wiring does not make a qubit architecture, there is readout, initialisation and measurement involved. Statements like "The present device of course does not extend P dots into the second dimension, but only a straightforward extension of the fabrication process used to make it would be needed to enable it." and "We believe our architecture is the most extensible of any experimentally published platform in silicon" are again not helpful if unsubstantiated, they only seem evangelical.

5) Throughout the response, I read statements where the weakness of one specific paper (out of the many the reviewers point out to demonstrate how far the field has progressed) is used to state

without any evidence that their system is capable of doing better, if only they had done so and demonstrated the experiment:

"There should be no doubt however, given our randomized benchmarking capability and fidelity, that our system is more than capable of executing a less complicated sequence like the ones in the compared papers."

I find the authors' stance in this matter quite unappealing and I am afraid I cannot take them at their word; if they had performed these "less complicated sequences" before the other (five?) groups mentioned across the two reviewer reports, no doubt the reviewers would have been as positive. If the authors believe they could perform a six-qubit experiment, or very easily demonstrate an extension into the second dimension of an array geometry, they should have done so. At the end of the day, regardless of future promise, their demonstration is of a two-qubit gate, and as far as I understand the role of publication in Nature is not to bring visibility to the future potential of a research field.

6) When the authors say "By "locally", we mean fields that change on the scale of the inter-dot spacing, since uniform magnetic fields by construction do not affect a DFS-encoded qubit, even if they fluctuate in time."

They mean a theoretical, or perfectly encoded DFS qubit, I believe.

7) I am not sure I appreciate the lesson about "magnetic noise" and "leakage", there was not any confusion, at least in my mind, about the nature of these terms, or if they are in competition with each other, only (a) how they are used in the paper, and (b) what is the top limiting factor in their fidelity.

I am glad the authors have clarified the usage of these terms in the paper. But, I am still unconvinced that mitigating the hyperfine via isotopic purification beyond the quantum well by just a few nm more would improve their fidelities much. It does not seem that the noise analysis is as straightforward as is claimed.

Actually, "Experimentally separating the total gate error into "encoded error," that is, error which occurs within the encoded subsystem, and "leakage error," that is, error which leaves spins in unencoded states, remains a challenge with two encoded qubits" is exactly the only kind of response I was asking for (indeed, it is ok that this is hard and has not been performed). It may not even be necessary, but if rigorous levels of noise analysis are claimed, then stating something like "We are quite confident about the amount of leakage vs. encoded error due to magnetic noise via simulation." is not ideal when the main text relies on this to make the case that isotopic purification can remove the problem.

At the end of the day the manuscript is interesting and many technical challenges for this particular type of spin qubit have been overcome. It also reports interesting and useful findings for others working in the field; whether it is suitable or not for Nature in terms of impact is better left up to the editors.

Referee #3 (Remarks to the Author):

In this revised manuscript, the authors tried to emphasize the novelty of the exchange-only encoded qubit scheme, which is quite different from conventional single-spin qubit schemes. This is the first experimental demonstration of the full universal gate set with exchange-only encoded qubits.

The scalability into a two-dimensional array of quantum dots using the SLEDGE architecture is now more clearly explained with an added figure in Fig.2(d).

SI section I.3 on crosstalk was also very helpful in addressing my previous questions about the parallel operations of exchange gates.

Overall, I think this revised manuscript made improvements to convey the relevance/importance of the exchange-only qubit approach.

I still recommend it be accepted and published in Nature.

Referee #4 (Remarks to the Author):

I am a new reviewer to this process, and I have read the materials. The question I have been asked to consider is whether I believe that the report of two-qubit gates between encoded qubits represents a very significant achievement in comparison to the state of the art in the field, in a way that could be argued to be comparable to the operation of six qubits in Philips et al., Nature 609, 919-924 (2022), and why.

Here are reasons that the paper of Weinstein et al., represents an achievement comparable to or beyond the paper of Philips et al..

1. The gate architecture used in Weinstein et al., has much better prospects for scalability than architectures used by other recent high-profile papers by academic groups. It is precisely the SLEDGE architecture that enables this, and the achievement reported by Weinstein et al., is indeed remarkable.
2. The qubits in use by Weinstein et al., involve a decoherence-free subsystem of a set of three spins. The paper of Weinstein et al., is the first time a two-qubit gate between such qubits has been demonstrated. In contrast, for example, the two-qubit gates implemented by Philips et al., were previously demonstrated by others. In this sense, the work of Weinstein et al., is a qualitative leap forward.
3. In their rebuttal, Weinstein et al., correctly indicate that measuring a concurrence or Bell-state fidelity (as was done in the paper of Philips et al.,) is not as strong as measuring a gate fidelity, as Weinstein et al., have done.
4. It is remarkable that the encoded qubit gate fidelities are as high as they are. Many exchange gates between the individual spins (each exchange gate corresponds to a single two-qubit gate in the device of Philips et al.,) are needed for a single two-qubit gate between encoded qubits. That such encoded gates work as well as they do is astonishing, and the authors indicate in the rebuttal that the error for each exchange gate is likely much smaller than what other groups have achieved.
5. The paper and work of Weinstein et al., do a really good job of emphasizing two critical issues for the spin-qubit and quantum-computing communities. These are the issues of scalable architectures

and choice of qubit encoding for spins. As Weinstein et al., point out in their rebuttal, these are essential issues for the community to sort out for continued progress.

Here are some reasons that the current paper might not represent a comparable achievement.

1. The SLEDGE architecture and potential for low SPAM errors have been reported in other papers by HRL.
2. It is true that while the six dots involved in the two qubits function remarkably well (arguably as well as or better than the dots in any other six-dot device reported in the literature, as the authors indicate), the six dots are configured as only two qubits, and those two qubits work about as well as other qubits in recent reports.

I think the fundamental issue is precisely what the authors mention at the end of their rebuttal. Their stated task is to convince the wider quantum computing community that EO DFS qubits are worthwhile to use and study. The paper of Weinstein et al., is a significant step in this direction.

Reviewer #5 paraphrased comments follow:

Reviewer #5 believes that the work by HRL certainly deserves publication in Nature. However, they believe that the overly technical presentation may have contributed to explain the reservations expressed by Reviewers #1 and #2. They believe a possible more accessible way to pitch your paper would be as follows:

The work realizes the first prototype of a quantum processor that relies on the Pauli exclusion principle, i.e., the exchange interaction between pair of single electron spins, to encode and process quantum information. Using 6 physical qubits, i.e., 6 electron spins in 6 quantum dots, they encode 2 logical qubits and demonstrate universal control as a first proof-of-concept. But this exchange-only quantum processor can do much more than that, e.g., it can shuttle quantum information around on a chip, or can be programmed for useful NISQ applications, i.e. a quantum simulators that harnesses exchange between fermions as a quantum resource that cannot be emulated with other platform (e.g. ion traps, superconducting qubits), as naturally and efficiently as spin qubits. Moreover, HRL's approach only requires electric signals which are delivered with the same hardware used to define the qubits, in contrast to the other state-of-the-art spin-based quantum processors in quantum dots, which require additional hardware (DC magnetic fields, AC magnetic fields, microwave pulses, spin-orbit coupling, magnetic field gradients). As importantly, HRL prototype is achieved on a semiconductor chip in a way that is truly manufacturable in existing microelectronic foundries i.e. the SLEDGE quantum dots architecture. This all-electrical, foundry-ready approach is extremely appealing for scaling spin qubits technology to the level required for turning this quantum technology into meaningful applications. To paraphrase what Michael Brookes from the New Scientist journal said in 2014: "Superconducting qubits are for those who like to play it safe, but spin qubits might very well overtake it during the next decade". Well, with HRL work, it's happening.

They also believe that this manuscript is as impressive, if not more, than the recent 6 qubit Delft paper. The key reasons are the invention of a new two-qubit gate and the subtractive manufacturing process that is industry ready. They believe with this work HRL moves ahead of Intel.

Their recommendation is to publish this work in Nature but making it more appealing to our broad audience. Most of their points are made in your work already, but they are buried in the text. Therefore, a more careful introduction, conclusion and outlook discussion would be very helpful in that respect, as opposed to the current focus on benchmarking.

REFEREE COMMENTS

We respond to the comments of Reviewers 1, 2, and 5 below:

Reviewer 1

Could the authors comment on the new Fig.2 d on the feasibility of maintaining the same interconnect footprint as one moves up in the metal layers of the SLEDGE architecture? Could the authors extend on how readout will happen, i.e. where the sensor will be located?

We prefer to not make overly-speculative comments about the specific manner that multi-layer SLEDGE devices would be routed. Such details would be more appropriate for a future paper demonstrating such a device. However, the main claim we do make about SLEDGE is that its BEOL is similar enough to commercial semiconductor technologies that the standard techniques used there should be applicable to SLEDGE as well. Regarding readout, we again view SLEDGE as a toolkit for building more advanced devices rather than a specific roadmap prescribing development. Simply put, there are lots of ways that readout could and will be done.

Despite these reservations, we see the value in addressing Reviewer 1's comment and have added the following phrase to the end of Fig 2's caption:

“Devices designed using this capability could feature multiple rails of coupled quantum dots and readout channels by leveraging the standard methods of BEOL signal interconnection used in the semiconductor microelectronics industry.”

Reviewer 2

I very much appreciated the added discussion of crosstalk, as I feel it strengthens the paper. However, the authors describe crosstalk as not a problem due to the high on-off nature of the exchange interaction, but then say that they were just unable to perform simultaneous operation in this flagship experiment, and had to compensate by staggering pulses slightly. Point 3 of Reviewer 1 also refers to this, I believe, and the answer to me is not satisfactory. They say on page 4 of the response that they perform “interleaving RB on one encoded qubit with quantum action on a second, showing no adverse effect of the parallel operation” as an argument for their low-crosstalk approach again; it is yet again unclear (even in the caption of the figure!) if “parallel” means “simultaneous” or if the pulses are again slightly staggered, because they then say in their response to point 4 of reviewer 1: “The distance at which simultaneous control pulses could safely be applied without meaningful error is somewhat larger than the present device, so we do not employ simultaneous control here.” and to me “However, we do not do that here, instead scheduling our control signals to assure that we never simultaneously engage two exchange interactions at once.” The reviewer clearly meant “parallel” in the sense of “simultaneous” here. I think this is dangerous, and the authors should be very clear at all times, since most people in the field would understand crosstalk during qubit control pulses to refer to simultaneous operation.

The issue of crosstalk in our technology is subtle and we appreciate the feedback from Reviewer 2 that our treatment was not as clear as it could be. Though we did discuss crosstalk extensively (and perhaps, confusingly) in our response letter, here we focus on the main text and supporting material itself. There are two main areas where the topic is discussed: the Extended Data Figure 2 and the “Crosstalk” section in Methods. We have added a clarifying sentence to the figure caption, saying “As discussed in Methods, this data serves as a validation of our approach for limiting crosstalk error, which includes strict avoidance of simultaneous exchange pulses and low susceptibility to spurious signals in the device “idle” configuration.” This is intended to emphasize that “simultaneous” operation (in the sense Reviewer 2 means it) was *not* evaluated here, and also to point the reader to the much more thorough discussion in the Methods.

We have also significantly revised the text in the Methods describing crosstalk. In particular, we have attempted to make more clear the distinction between interleaved “parallel” operation and truly simultaneous operation, and the underlying physics reasons the two are very different. We have also hopefully made the prospects for actual simultaneous operation in our device technology more clear

(e.g. slightly bigger devices are needed, but it should be ok).

In reviewing these changes, we also noticed that the references in the Main Text to the Extended Data figures were incorrect, which certainly didn't help to keep the story clean. Those errors have also been corrected. We also updated the SPAM fidelity in the Extended Data figure 2 (as the earlier number pertained to data not included here) and now correctly report the corresponding number for the data shown as $96.0 \pm 0.1\%$.

Reviewer 5

(editor) With respect to the suggestion of Reviewer 5, I wouldn't ask you to overhaul your manuscript at this time nor to necessarily directly implement the suggestions of Reviewer 5, but I think they are helpful suggestions. As you will see below I do agree that at least your abstract fails to communicate in an accessible way the key results of your work and why they represent a significant advance in technology. We therefore will ask you to revise your abstract, but if you also could streamline and simplify your introductions and conclusions that would be considerably helpful in making your work as accessible as possible to a broader audience. I am happy to give you feedback during the process if you'd like me to.

The work realizes the first prototype of a quantum processor that relies on the Pauli exclusion principle, i.e., the exchange interaction between pair of single electron spins, to encode and process quantum information. Using 6 physical qubits, i.e., 6 electron spins in 6 quantum dots, they encode 2 logical qubits and demonstrate universal control as a first proof-of-concept. But this exchange-only quantum processor can do much more than that, e.g., it can shuttle quantum information around on a chip, or can be programmed for useful NISQ applications, i.e. a quantum simulators that harnesses exchange between fermions as a quantum resource that cannot be emulated with other platform (e.g. ion traps, superconducting qubits), as naturally and efficiently as spin qubits. Moreover, HRL's approach only requires electric signals which are delivered with the same hardware used to define the qubits, in contrast to the other state-of-the-art spin-based quantum processors in quantum dots, which require additional hardware (DC magnetic fields, AC magnetic fields, microwave pulses, spin-orbit coupling, magnetic field gradients). As importantly, HRL prototype is achieved on a semiconductor chip in a way that is truly manufacturable in existing microelectronic foundries i.e. the SLEDGE quantum dots architecture. This all-electrical, foundry-ready approach is extremely appealing for scaling spin qubits technology to the level required for turning this quantum technology into meaningful applications.

We appreciate the thought that Reviewer 5 has clearly given to the presentation of our result and his or her recommendations show a deep level of understanding of its significance. While we agree with you that a second major revision to the manuscript is not practical, they do offer helpful recommendations that are not

too difficult to implement in part:

- As we discuss in the “summary paragraph” section below, we have removed some more technical language from the abstract, following your more specific recommendations. We attempt to hit at some level all of the pertinent points listed by reviewer 5, but without using the technical language reviewer 5 has used. For example, “requires electric signals which are delivered with the same hardware used to define the qubits” is a great way to clarify the advantage to experts looking at quantum dots, but perhaps not to the general reader, as significant explanation of what is meant by “define the quantum dots” would be needed. So we have done our best to capture the spirit of each comment while preserving accessibility.
- We have increased the emphasis on the manufacturability of the SLEDGE device following Reviewer 5’s recommendation (e.g. our “foundry-ready approach”).
- We have also added “all-electrical” control to the last introductory sentence.
- We have added a statement relating exchange to the Pauli exclusion principle, which is a more commonly known phenomena.
- Finally, their note that our system could be used as “quantum simulators that harnesses exchange between fermions as a quantum resource that cannot be emulated with other platform” is an interesting one, and an avenue of research that we had not much addressed in our prior submission. We have added a sentence to the conclusion acknowledging the possible application to analog simulation. For this we cite again the 2017 Nature paper from Delft that implemented a Fermi-Hubbard simulation in a comparable system (which could be done at larger scale in our platform), as well as a very interesting new result, still only on the arXiv, from a different group at Delft which effectively does exchange-only control using Ge holes in a four-dot device and casts the result as a molecular simulation.